Workshop at the 6th Symposium on Advances in Approximate Bayesian Inference (non-archival), 2024 1–28

# Variational Linearized Laplace Approximation for Bayesian Deep Learning

**Luis A. Ortega**                                          luis.ortega@uam.es
*Universidad Autónoma de Madrid*

**Simón Rodriguez Santana**                    srsantana@icai.comillas.edu
*Universidad Pontificia Comillas*

**Daniel Hernández-Lobato**                      daniel.hernandez@uam.es
*Universidad Autónoma de Madrid*

## Abstract

The Linearized Laplace Approximation (LLA) has been recently used to perform uncertainty estimation on the predictions of pre-trained deep neural networks (DNNs). However, its widespread application is hindered by significant computational costs, particularly in scenarios with a large number of training points or DNN parameters. Consequently, additional approximations of LLA, such as Kronecker-factored or diagonal approximate GGN matrices, are employed, potentially compromising the model's performance. To address these challenges, we propose a new method for approximating LLA using a variational sparse Gaussian Process (GP). Our method is based on the dual RKHS formulation of GPs and retains as the predictive mean the output of the original DNN. Furthermore, it allows for efficient stochastic optimization, which results in sub-linear training time in the size of the training dataset. Specifically, its training cost is independent of the number of training points. We compare our proposed method against accelerated LLA (ELLA), which relies on the Nyström approximation, as well as other LLA variants employing the sample-then-optimize principle. Experimental results show that our method outperforms these already existing efficient variants of LLA, both in terms of the quality of the predictive distribution and in terms of total computational time.

## 1. Introduction

Deep neural networks (DNNs) have gained widespread popularity for addressing pattern recognition problems due to their state-of-the-art performance in predicting target values from a set of input attributes (He et al., 2016; Vaswani et al., 2017). Despite this great success, DNNs exhibit limitations when computing a predictive distribution that accounts for the confidence in the predictions. Specifically, DNNs result in weak calibration (Guo et al., 2017) and in poor reasoning regarding model uncertainty (Blundell et al., 2015). These issues become particularly critical in risk-sensitive situations like autonomous driving (Kendall and Gal, 2017) and healthcare systems (Leibig et al., 2017) among others.

The Laplace Approximation (LA) leverages the maximum a posteriori (MAP) solution, attainable via back-propagation, providing a Gaussian posterior approximation where the pre-trained optimal solution furnishes the mean. This resembles a pre-training step followed by fine-tuning, a common practice in deep learning (Daxberger et al., 2021a). LA's weakness is the necessity of computing the Hessian at the MAP, which becomes prohibitive for large

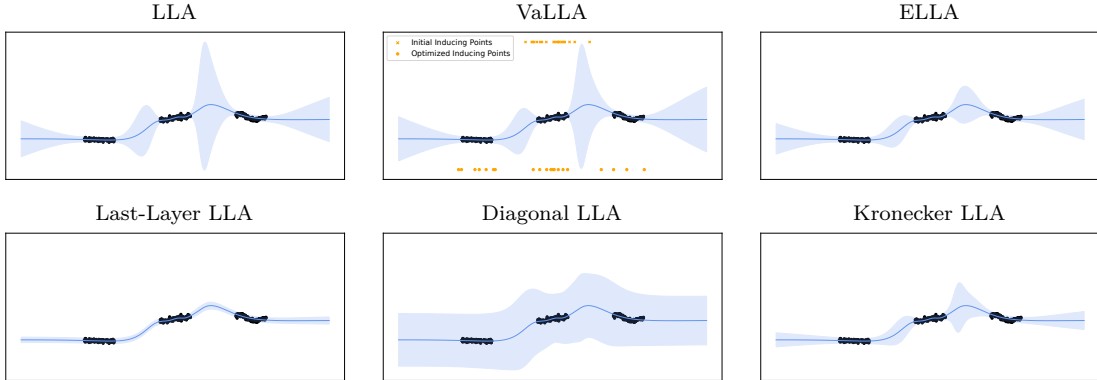

Figure 1: Predictive distribution (mean in blue and shaded two times the standard deviation) on a toy 1D regression dataset with a 2 hidden layer MLP with 50 units trained using back-propagation. The predictive distribution of VaLLA is on par with or better than other approximations (last layer, diagonal and Kronecker factorization) and other methods (ELLA). The optimal values for the noise and the Gaussian prior are optimized by maximizing the marginal log likelihood estimate of LLA. VaLLA and ELLA use the optimal values found by LLA. VaLLA uses 20 inducing points for the predictive variances. ELLA uses 20 random locations and 20 features.

DNNs. To simplify this, the Hessian is often approximated using the generalized Gauss-Newton (GGN) matrix (Martens and Grosse, 2015). This approach aligns with the fact that the GGN Hessian estimate is exact when applying LA to a linearized DNN. This method is referred to as Linearized LA (LLA) (Immer et al., 2021). However, in the context of large DNNs, further approximations on top of GGN are required.

In this work, we broaden LLA's usage for DNNs to cases with a substantial number of parameters and training instances. We reinterpret the LLA predictive distribution as that of a Gaussian Process (GP) (Khan et al., 2019), and use a sparse variational GP based on inducing points to approximate it (Titsias, 2009). Standard sparse GPs often alter the predictive mean and hence may deteriorate the accuracy of the pre-trained DNN, which is assumed to have minimal generalization error. To avoid this, we use a dual representation of the GP in the Reproducing Kernel Hilbert Space (RKHS). This enables the sparse approximation to be used only in the computation of the predictive variances of the GP. The result is a sparse GP approximation where the predictive mean coincides with the predictions of the pre-trained DNN, without introducing additional prediction error.

We call our method Variational LLA (VaLLA) and conduct a series of experiments to compare its performance and computational cost with that of related methods from the literature. Our comparisons include (i) a method that uses Kronecker and diagonal approximations of the GGN matrix, (ii) a method that uses a Nyström approximation of that matrix (ELLA) (Deng et al., 2022), and (iii) a method that relies on generating samples from LLA's posterior distribution using the sample-then-optimize principle (Antorán et al.,

2023). VaLLA's performance is comparable or better than that of these other methods and is obtained at a smaller cost. Specifically, VaLLA adopts mini-batch training, resulting in sub-linear cost w.r.t. the training set size. Furthermore, VaLLA often generates predictive distributions that closely resemble those of LLA. Figure 1 shows this in a simple 1-D toy regression problem.

## 2. Background

Consider the task of inferring an unknown function $f : \mathbb{R}^D \to \mathbb{R}$ based on noisy observations $\mathbf{y} = (y_1, \ldots, y_N)^{\mathrm{T}}$ at corresponding locations $\mathbf{X} = (\mathbf{x}_1, \ldots, \mathbf{x}_N)$. Deep learning (DL) defines a Neural Network $g : \mathbb{R}^D \times \mathbb{R}^P \to \mathbb{R}$ with $P$ parameters so that $\exists \boldsymbol{\theta}^\star \in \mathbb{R}^P$ $s.t.$ $f(\cdot) \approx g(\cdot, \boldsymbol{\theta}^\star)$.

The Laplace approximation (LA) builds a Gaussian approximate posterior (Mackay, 1992). This takes the form of $q(\boldsymbol{\theta}) = \mathcal{N}(\boldsymbol{\theta}|\hat{\boldsymbol{\theta}}, \boldsymbol{\Sigma})$, where $\hat{\boldsymbol{\theta}}$ denotes the MAP solution, $i.e.$, $\hat{\boldsymbol{\theta}} = \arg\max_{\boldsymbol{\theta}} \log p(\mathbf{y}|\boldsymbol{\theta}) + \log p(\boldsymbol{\theta})$, and $\boldsymbol{\Sigma}$ is the inverse of the negative Hessian of the log posterior, $i.e.$,$\boldsymbol{\Sigma}^{-1} = -\nabla^2_{\boldsymbol{\theta}\boldsymbol{\theta}} \left[\log p(\mathbf{y}|\boldsymbol{\theta}) + \log p(\boldsymbol{\theta})\right]|_{\boldsymbol{\theta}=\hat{\boldsymbol{\theta}}}$. Often, an isotropic Gaussian prior $p(\boldsymbol{\theta}) = \mathcal{N}(\boldsymbol{\theta}|\mathbf{0}, \sigma_0^2 \boldsymbol{I}_P)$ is considered. Due to the intractability of the Hessian in large DNNs, it is common to approximate it with the generalized Gauss-Newton (GGN) matrix (Immer et al., 2021):

$$\boldsymbol{\Sigma}^{-1} \approx \sum_{n=1}^N J_{\hat{\boldsymbol{\theta}}}(\mathbf{x}_n)^{\mathrm{T}} \Lambda(\mathbf{x}_n, y_n) J_{\hat{\boldsymbol{\theta}}}(\mathbf{x}_n) + \frac{1}{\sigma_0^2} \boldsymbol{I}_P \,, \tag{1}$$

where $J_{\hat{\boldsymbol{\theta}}}(\mathbf{x}_n) = \nabla_{\boldsymbol{\theta}} \, g(\mathbf{x}_n, \boldsymbol{\theta})|_{\boldsymbol{\theta}=\hat{\boldsymbol{\theta}}}$ and $\Lambda(\mathbf{x}_n, y_n) = -\nabla^2_{\mathbf{g}\mathbf{g}} \log p(y_n|\mathbf{g})|_{\mathbf{g}=g_{\hat{\boldsymbol{\theta}}}(\mathbf{x}_n, \boldsymbol{\theta})}$. The GGN matrix is guaranteed to be positive definite, which means that $\hat{\boldsymbol{\theta}}$ need not be a maximum of the log posterior. It can be, $e.g.$, any solution found by early-stopping back-propagation.

The earlier formulation of LA suffers from underfitting (Lawrence, 2001). This is attributed to the fact that the GGN approximation is the true Hessian matrix of the linearized DNN $g^{\mathrm{lin}}_{\hat{\boldsymbol{\theta}}}(\mathbf{x}, \boldsymbol{\theta}) := g(\mathbf{x}, \hat{\boldsymbol{\theta}}) + J_{\hat{\boldsymbol{\theta}}}(\mathbf{x}_n)(\boldsymbol{\theta} - \hat{\boldsymbol{\theta}})$ (Immer et al., 2021). This implies a shift between posterior inference and predictions that can be mitigated by also predicting using the linearized model:

$$p_{\mathrm{LLA}}(y^\star|\mathbf{x}^\star, \mathbf{y}) = \mathbb{E}_{q(\boldsymbol{\theta})} \left[ p(y^\star|g^{\mathrm{lin}}_{\hat{\boldsymbol{\theta}}}(\mathbf{x}^\star, \boldsymbol{\theta})) \right] \,. \tag{2}$$

This method is known as the linearized LA (LLA). Despite these approximations, LLA still requires the inversion of $\boldsymbol{\Sigma}^{-1}$ which scales cubically with the number of parameters of the DNN. A dual formulation of LLA as a Gaussian Process (GP), described in the next section, scales with cubic cost in the number of training points.

### 2.1. Gaussian Process (GP) Interpretation of LLA

Using the approximate posterior from LLA, $i.e.$, $q(\boldsymbol{\theta}) = \mathcal{N}(\boldsymbol{\theta}|\hat{\boldsymbol{\theta}}, \boldsymbol{\Sigma})$, we also obtain a GP for prediction. The mean and covariance functions, providing the predictive mean and variances of the LLA approximation, are in this case $m(\mathbf{x}) = g(\mathbf{x}, \hat{\boldsymbol{\theta}})$, and $K(\mathbf{x}, \mathbf{x}') = J_{\hat{\boldsymbol{\theta}}}(\mathbf{x})^{\mathrm{T}} \boldsymbol{\Sigma} J_{\hat{\boldsymbol{\theta}}}(\mathbf{x}')$, respectively. Using the Woodbury formula on $\boldsymbol{\Sigma}$ and defining $\kappa(\mathbf{x}, \mathbf{x}') = \sigma_0^2 J_{\hat{\boldsymbol{\theta}}}(\mathbf{x})^{\mathrm{T}} J_{\hat{\boldsymbol{\theta}}}(\mathbf{x}')$ as the (scaled) Neural Tangent Kernel ($i.e.$, prior covariance function) of the GP (Immer et al., 2021), and $\mathbf{Q} = \Lambda_{\mathbf{X},\mathbf{y}}^{-1} + \kappa(\mathbf{X}, \mathbf{X})$, the covariance function of the GP takes the expression

$$K(\mathbf{x}, \mathbf{x}') = \kappa(\mathbf{x}, \mathbf{x}') - \kappa(\mathbf{x}, \mathbf{X}) \mathbf{Q}^{-1} \kappa(\mathbf{X}, \mathbf{x}') \,. \tag{3}$$

This allows us to interpret the LLA approximate predictive distribution as a posterior GP in function space, with prior covariance function given by $\kappa(\cdot, \cdot)$. The bottleneck of this interpretation is the evaluation of $\mathbf{Q}^{-1}$, with $\mathcal{O}(N^3 + N^2 P)$ cost. In the case of classification problems with $C$ classes, $\kappa(\mathbf{X}, \mathbf{X})$ and $\mathbf{Q}$ have size $NC \times NC$, thus the cost is also cubic in the number of classes $C$ or DNN outputs.

## 2.2. Dual formulation of Gaussian Processes in RKHS

A Reproducing Kernel Hilbert Space (RKHS) $\mathcal{H}$ is a Hilbert space of functions with the reproducing property: $\forall \mathbf{x} \in \mathcal{X} \; \exists \phi_{\mathbf{x}} \in \mathcal{H}$ such that $\forall f \in \mathcal{H}, f(\mathbf{x}) = \langle \phi_{\mathbf{x}}, f \rangle$. In general, $\mathcal{H}$ can be infinite-dimensional and approximate continuous functions on a compact set.

A zero-mean prior GP with posterior mean and covariance functions $m(\cdot)$ and $K(\cdot, \cdot)$, respectively, has a dual representation in a RKHS (Cheng and Boots, 2016). There exists $\mu \in \mathcal{H}$ and a linear semi-definite positive operator $\Sigma : \mathcal{H} \to \mathcal{H}$ s.t. $\forall \mathbf{x}, \mathbf{x}' \in \mathcal{X}, \exists \phi_{\mathbf{x}}, \phi_{\mathbf{x}'}$ verifying $m(\mathbf{x}) = \langle \phi_{\mathbf{x}}, \mu \rangle$ and $K(\mathbf{x}, \mathbf{x}') = \langle \phi_{\mathbf{x}}, \Sigma(\phi_{\mathbf{x}'}) \rangle$.

## 3. Variational LLA (VaLLA)

Variational sparse GPs approximate the GP posterior using a GP parameterized by $M$ inducing points $\mathbf{Z}$, each in $\mathbb{R}^D$, and associated process values $\mathbf{u} = f(\mathbf{Z})$ (Titsias, 2009), $p(\mathbf{f}, \mathbf{u}|\mathbf{y}) \approx q(\mathbf{f}, \mathbf{u}) = p(\mathbf{f}|\mathbf{u})q(\mathbf{u})$, where $q(\mathbf{u}) = \mathcal{N}(\mathbf{u}|\hat{\boldsymbol{m}}, \hat{\boldsymbol{S}})$, $\mathbf{f} = f(\mathbf{X})$ and $p(\mathbf{f}|\mathbf{u})$ is fixed. The approximate distribution $q(\mathbf{u})$ is obtained by minimizing the KL-divergence $\text{KL}\left(q(\mathbf{f}, \mathbf{u}) \mid p(\mathbf{f}, \mathbf{u}|\mathbf{y})\right)$. In practice, the minimization problem is transformed into the maximization of the lower bound of the log-marginal likelihood

$$\log p(\mathbf{y}) \geq \max_{\mathbf{Z}, \hat{\boldsymbol{m}}, \hat{\boldsymbol{S}}} \int_{\mathbf{f}, \mathbf{u}} q(\mathbf{f}, \mathbf{u}) \log \frac{p(\mathbf{y}|\mathbf{f})p(\mathbf{f}|\mathbf{u})p(\mathbf{u})}{q(\mathbf{f}, \mathbf{u})} \; , \tag{4}$$

with cost $\mathcal{O}(NM^2 + M^3)$ due to $q(\mathbf{f}, \mathbf{u}) = p(\mathbf{f}|\mathbf{u})q(\mathbf{u})$.

**Theorem 1** *(Cheng and Boots, 2016) Using a sparse GP approximation with $q(\mathbf{f}, \mathbf{u}) = p(\mathbf{f}|\mathbf{u})q(\mathbf{u})$ is equivalent to restricting the mean and covariance functions of the dual representation in the RKHS to $\tilde{\mu} = \Phi_{\mathbf{Z}}(\boldsymbol{a})$ and $\tilde{\Sigma} = I + \Phi_{\mathbf{Z}}\boldsymbol{A}\Phi_{\mathbf{Z}}^T$. Where the functional $\Phi_{\mathbf{Z}} : \mathbb{R}^M \to \mathcal{H}$ defines a linear combination of basis functions as $\Phi_{\mathbf{Z}}(\boldsymbol{a}) = \sum_{m=1}^M a_m \phi_{\mathbf{z}_m}$, with $\boldsymbol{a} = (a_1, \ldots, a_M) \in \mathbb{R}^M$ and the functional $\Phi_{\mathbf{Z}}\boldsymbol{A}\Phi_{\mathbf{Z}}^T = \sum_{i=1}^M \sum_{j=1}^M \phi_{\mathbf{z}_i} A_{i,j} \phi_{\mathbf{z}_j}^T$, defines a quadratic expression where $\boldsymbol{A} \in \mathbb{R}^{M \times M}$ such that $\tilde{\Sigma} \geq 0$. Proof in Appendix A.*

Theorem 1 indicates that the algorithm of Titsias (2009) optimizes a variational Gaussian measure where $\tilde{\mu}$ and $\tilde{\Sigma}$ are parameterized by a function basis $\{\phi_{\mathbf{z}} \in \mathcal{H} \mid \mathbf{z} \in \mathbf{Z}\}$. Cheng and Boots (2017) propose to generalize this so that each of the linear operators is optimized using different bases (sets of inducing points). Let $\mathbf{Z}_{\alpha}$ and $\mathbf{Z}_{\beta}$ be two sets of inducing points for the mean and the variance, respectively. The parameterization of Cheng and Boots (2017) is $\tilde{\mu} = \Phi_{\mathbf{Z}_{\alpha}}(\boldsymbol{a})$ and $\tilde{\Sigma} = (I + \Phi_{\mathbf{Z}_{\beta}}\boldsymbol{A}\Phi_{\mathbf{Z}_{\beta}}^T)^{-1}$. Where $\Phi_{\mathbf{Z}_{\alpha}} : \mathbb{R}^{M_{\alpha}} \to \mathcal{H}$ and $\Phi_{\mathbf{Z}_{\beta}} : \mathbb{R}^{M_{\beta}} \to \mathcal{H}$ are defined as $\Phi_{\mathbf{Z}}$ using $\mathbf{Z}_{\alpha}$ and $\mathbf{Z}_{\beta}$, respectively. Now, there are two sets of inducing points, $M_{\alpha}$ for the mean and $M_{\beta}$ for the covariances, respectively. This

parameterization is a generalization and cannot be obtained using the approach of Titsias (2009). Here, $q$ must be found by optimizing Gaussian measures (Cheng and Boots, 2016):

$$\max_{q(f)} \mathcal{L}(q(f)) = \max_{q(f)} \mathbb{E}_q \left[ \log p(\mathbf{y}|f) \right] - \mathrm{KL}\left( q \mid p \right) , \tag{5}$$

where $\mathrm{KL}\left( q \mid p \right) = \frac{1}{2}\boldsymbol{a}^T \boldsymbol{K}_\alpha \boldsymbol{a} + \frac{1}{2}\log|\boldsymbol{I} + \boldsymbol{K}_\beta \boldsymbol{A}| - \frac{1}{2}\mathrm{tr}\left( \boldsymbol{K}_\beta(\boldsymbol{A}^{-1} + \boldsymbol{K}_\beta)^{-1} \right)$; with $\boldsymbol{K}_\alpha$ and $\boldsymbol{K}_\beta$ matrices with the prior covariances among $f(\mathbf{Z}_\alpha)$ and $f(\mathbf{Z}_\beta)$, respectively. In practice, the objective leads to null variances and $\alpha$-divergences with Early-Stopping are used instead of the standard ELBO (see Appendix D for further information).

### 3.1. Using Decoupled SGP and LLA

We use the decoupled reparameterization of sparse GPs to establish a model where the mean of the approximated posterior distribution is fixed to a pre-trained MAP solution. We denote this method as *variational LLA* (VaLLA).

**Proposition 2** *If $g(\cdot, \hat{\boldsymbol{\theta}}) \in \mathcal{H}$, then $\forall \epsilon > 0$ exists a set of $M_\alpha$ inducing points $\mathbf{Z}_\alpha$ and a collection of scalar values $\boldsymbol{a} \in \mathbb{R}^{M_\alpha}$ such that the dual representation of the sparse Gaussian process defined by $\tilde{\mu} = \Phi_{\mathbf{Z}_\alpha}(\boldsymbol{a})$ and $\tilde{\Sigma} = (I + \Phi_{\mathbf{Z}_\beta} \boldsymbol{A} \Phi_{\mathbf{Z}_\beta}^T)^{-1}$ corresponds to a GP posterior approximation with mean $m^\star(\mathbf{x}) = h_\epsilon(\mathbf{x})$ and covariance function defined as*

$$K^\star(\mathbf{x}, \mathbf{x}') = K(\mathbf{x}, \mathbf{x}') - K_{\mathbf{x}, \mathbf{Z}_\beta}(\boldsymbol{A}^{-1} + \boldsymbol{K}_\beta)^{-1} K_{\mathbf{Z}_\beta, \mathbf{x}'} , \tag{6}$$

*where $\mathbf{Z}_\beta$ is a set of $M_\beta$ inducing points, $\boldsymbol{A} \in \mathbb{R}^{M_\beta \times M_\beta}$, $K_{\mathbf{x}, \mathbf{Z}_\beta}$ is a vector with the covariances between $f(\mathbf{x})$ and $f(\mathbf{Z}_\beta)$, and $h_\epsilon$ verifies $d_\mathcal{H}(g(\cdot, \hat{\boldsymbol{\theta}}), h_\epsilon) \leq \epsilon$, with $d_\mathcal{H}(\cdot, \cdot)$ the distance in the RKHS. Proof in Appendix A.*

Proposition 2 implies that if $g(\cdot, \hat{\boldsymbol{\theta}}) \in \mathcal{H}$ we can find values for $\boldsymbol{a}$ and inducing points for the mean $\mathbf{Z}_\alpha$ s.t. $d_\mathcal{H}(g(\cdot, \hat{\boldsymbol{\theta}}), h_\epsilon)$ can be made as small as desired. For sufficiently small $\epsilon$, $h_\epsilon(\cdot) \approx g(\cdot, \hat{\boldsymbol{\theta}})$, and $g(\cdot, \hat{\boldsymbol{\theta}})$ can be used for prediction instead of $h_\epsilon(\mathbf{x})$. Thus, there is no need to optimize $\boldsymbol{a}$ and $\mathbf{Z}_\alpha$ in (5), and the posterior distribution of VaLLA uses $g(\cdot, \hat{\boldsymbol{\theta}})$ as its mean function. The optimal parameters $\mathbf{Z}_\beta$ and $\boldsymbol{A}$ can be found by optimizing (5) with $\boldsymbol{a}$ and $\mathbf{Z}_\alpha$ held constant. From the following proposition, computing the optimal value of $\mathbf{A}$ has cost $\mathcal{O}(N M_\beta^2 + M_\beta^3)$.

**Proposition 3** *The value of $\boldsymbol{A}$ in Proposition 2 that minimizes Equation (5) is $\boldsymbol{A} = \frac{1}{\sigma^2}\boldsymbol{K}_\beta^{-1}\boldsymbol{K}_{\boldsymbol{Z}_\beta, \boldsymbol{X}}\boldsymbol{K}_{\boldsymbol{X}, \boldsymbol{Z}_\beta}\boldsymbol{K}_\beta^{-1}$ where $\sigma^2$ is the noise variance and $\boldsymbol{K}_{\boldsymbol{X}, \boldsymbol{Z}_\beta}$ is a matrix with the prior covariances between $f(\mathbf{X})$ and $f(\mathbf{Z}_\beta)$. If $\boldsymbol{Z}_\beta = \mathbf{X}$, the covariance function of the predictive distribution in (6) is equal to that of the full GP. Proof in Appendix A.*

### 3.2. Limitations of VaLLA

VaLLA is limited by three factors: (i) Computing the predictive distribution at each training iteration involves inverting $\mathbf{A}^{-1} + K_{\mathbf{Z}_\beta}$ in (6), with cubic cost in the number of inducing points $M_\beta$. Therefore, VaLLA cannot accommodate a very large number of inducing points. (ii) The minimization objective requires a validation set and the use of early-stopping to

| Method | ResNet-20 | | | ResNet-32 | | | ResNet-44 | | | ResNet-56 | | | Mean Rank |
|---|---|---|---|---|---|---|---|---|---|---|---|---|---|
| | ACC | NLL | ECE | ACC | NLL | ECE | ACC | NLL | ECE | ACC | NLL | ECE | |
| MAP | **92.6** | 0.282 | 0.039 | **93.5** | 0.292 | 0.041 | **94.0** | 0.275 | 0.039 | **94.4** | 0.252 | 0.037 | – |
| MF-VI | **92.7** | **0.231** | 0.016 | **93.5** | 0.222 | 0.020 | 93.9 | 0.206 | 0.018 | **94.4** | 0.188 | 0.016 | – |
| LLA Diag | 92.2 | 0.728 | 0.404 | **92.7** | 0.755 | 0.430 | 92.8 | 0.778 | 0.445 | **92.9** | 0.843 | 0.480 | – |
| LLA KFAC | 92.0 | 0.852 | 0.467 | 91.8 | 1.027 | 0.547 | 91.4 | 1.091 | 0.566 | 89.8 | 1.174 | 0.579 | – |
| LLA* | **92.6** | 0.269 | 0.034 | **93.5** | 0.259 | 0.033 | **94.0** | 0.237 | 0.028 | **94.4** | 0.213 | 0.022 | – |
| LLA* KFAC | **92.6** | 0.271 | 0.035 | **93.5** | 0.260 | 0.033 | **94.0** | 0.232 | 0.028 | **94.4** | 0.202 | 0.024 | – |
| ELLA | 92.5 | 0.233 | 0.009 | **93.5** | **0.215** | **0.008** | 93.9 | 0.204 | **0.007** | **94.4** | 0.187 | **0.007** | 2.375 |
| Sampled LLA | 92.5 | **0.231** | **0.006** | **93.5** | 0.217 | **0.008** | **94.0** | **0.200** | **0.007** | **94.4** | **0.185** | 0.015 | **2.000** |
| VaLLA | **92.6** | **0.228** | **0.007** | **93.5** | **0.211** | **0.007** | **94.0** | **0.198** | **0.008** | **94.4** | **0.183** | **0.009** | **1.375** |

Table 1: Results on CIFAR10. ACC, NLL and ECE are computed using Monte-Carlo estimation. Best value highlighted in **purple** and second to best in **teal**. Sampled LLA uses 64 samples. ELLA uses $M = 2000$ points and $K = 20$. Average results over 5 different random seeds (standard deviations $< 10^{-4}$ in all cases and omitted). * for Last Layer LLA. Mean rank only considers both NLL and ECE.

effectively optimize the prior variance $\sigma_0^2$, thus further increasing training time. (iii) Mini-batch optimization involves evaluating $K_{\mathbf{x},\mathbf{Z}_\beta} \ \forall \mathbf{x} \in \mathcal{B}$ and $\boldsymbol{K}_\beta$. Hence, we require efficient evaluation of the (scaled) Neural Tangent Kernel, $\kappa(\cdot,\cdot) = \sigma^2 J_{\hat{\boldsymbol{\theta}}}(\cdot)^{\mathrm{T}} J_{\hat{\boldsymbol{\theta}}}(\cdot)$ and its gradients to find $\mathbf{Z}_\beta$. While there are libraries that use structure in the derivatives for the efficient computation of $\kappa(\cdot,\cdot)$, these are limited to a few DNN models (Novak et al., 2022). A simple but inefficient approach to evaluate $\kappa(\cdot,\cdot)$ involves computing and storing all full Jacobians in memory, for each mini-batch instance and inducing point. This is tractable in our problems, but makes VaLLA infeasible in very large problems, *e.g.*, ImageNet. Appendix H.3 shows a very efficient *layer-by-layer* method to get $K_{\mathbf{x},\mathbf{Z}_\beta} \ \forall \mathbf{x} \in \mathcal{B}$ and $\boldsymbol{K}_\beta$. However, this requires computing each layer's contribution to the Jacobian at hand, which is difficult for large and complex DNNs.

## 4. Experiments

We compare VaLLA with other methods using the LLA implementation by Daxberger et al. (2021a) on various ResNet models trained on Cifar10 data. VaLLA uses a batch size of 100. We extract the results for other methods from Deng et al. (2022) and use the same DNN for VaLLA. The corresponding pre-trained models are those of Deng et al. (2022) (accessible here). Table 1 shows ACC, NLL and ECE for each method, including LLA variants and a mean-field VI approach (Deng and Zhu, 2023). We compare with Antorán et al. (2023)'s approach to retrieve samples from the true GP posterior defined by LLA. Since we use the same pre-trained models, the results of all other methods are consistent with those reported by Deng et al. (2022). VaLLA with $M_\beta = 100$ outperforms other methods in most cases, always being either the best or second-best method.

Appendix G contains further experimentation of the proposed method. More precisely, regression experiments in Year, Airline and Taxi datasets are reported; along with further classification results in MNIST and FMNIST with OOD detection.

## 5. Conclusions

We introduced VaLLA, a method derived from the formulation of a generalized sparse GP that offers the flexibility to fix the predictive mean to any desired function in the RKHS. VaLLA excels in computing error bars for pre-trained DNNs with a vast number of parameters on extensive datasets, handling even millions of training instances. VaLLA's applicability spans both regression and classification problems, showcasing costs independent of the number of training points $N$. In comparison, the Nyström approximation by Deng et al. (2022) incurs a linear cost in $N$, unless early-stopping is employed. Furthermore, VaLLA surpasses the sample-then-optimize method of Antorán et al. (2023) in terms of speed, while also providing predictive distributions robust to input corruptions. In essence, VaLLA stands out by delivering robust predictive distributions akin to LLA, all while maintaining noteworthy computational efficiency.

## Acknowledgments

Authors gratefully acknowledge the use of the facilities of Centro de Computacion Cientifica (CCC) at Universidad Autónoma de Madrid. The authors acknowledge financial support from project PID2022-139856NB-I00 funded by MCIN/ AEI / 10.13039/501100011033 / FEDER, UE and project PID2019-106827GB-I00 / AEI / 10.13039/501100011033 and from the Autonomous Community of Madrid (ELLIS Unit Madrid). The authors also acknowledge financial support from project TED2021-131530B-I00, funded by MCIN/AEI /10.13039/501100011033 and by the European Union NextGenerationEU PRTR.

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

## Appendix A. Proofs

**Theorem 1** *([Cheng and Boots, 2016](#)) Using a sparse GP approximation with $q(\mathbf{f}, \mathbf{u}) = p(\mathbf{f}|\mathbf{u})q(\mathbf{u})$ is equivalent to restricting the mean and covariance functions of the dual representation in the RKHS to $\tilde{\mu} = \Phi_{\mathbf{Z}}(\boldsymbol{a})$ and $\tilde{\Sigma} = I + \Phi_{\mathbf{Z}}\boldsymbol{A}\Phi_{\mathbf{Z}}^T$. Where the functional $\Phi_{\mathbf{Z}} : \mathbb{R}^M \to \mathcal{H}$ defines a linear combination of basis functions as $\Phi_{\mathbf{Z}}(\boldsymbol{a}) = \sum_{m=1}^M a_m \phi_{\mathbf{z}_m}$, with $\boldsymbol{a} = (a_1, \ldots, a_M) \in \mathbb{R}^M$ and the functional $\Phi_{\mathbf{Z}}\boldsymbol{A}\Phi_{\mathbf{Z}}^T = \sum_{i=1}^M \sum_{j=1}^M \phi_{\mathbf{z}_i} A_{i,j} \phi_{\mathbf{z}_j}^T$, defines a quadratic expression where $\boldsymbol{A} \in \mathbb{R}^{M \times M}$ such that $\tilde{\Sigma} \geq 0$.*

**Proof** First of all, notice that $\Phi_{\mathbf{Z}} = \left(\phi_{\mathbf{z}_1}, \cdots, \phi_{\mathbf{z}_M}\right) \in \mathcal{H}^M$, leading to

$$K_{\mathbf{Z}} := \Phi_{\mathbf{Z}}\Phi_{\mathbf{Z}}^T = \begin{pmatrix} \langle \phi_{\mathbf{z}_1}, \phi_{\mathbf{z}_1} \rangle & \langle \phi_{\mathbf{z}_1}, \phi_{\mathbf{z}_2} \rangle & \cdots & \langle \phi_{\mathbf{z}_1}, \phi_{\mathbf{z}_M} \rangle \\ \vdots & \vdots & & \vdots \\ \langle \phi_{\mathbf{z}_M}, \phi_{\mathbf{z}_1} \rangle & \langle \phi_{\mathbf{z}_M}, \phi_{\mathbf{z}_2} \rangle & \cdots & \langle \phi_{\mathbf{z}_M}, \phi_{\mathbf{z}_M} \rangle \end{pmatrix} \in \mathbb{R}^{M \times M}. \tag{7}$$

Furthermore, $K_{\mathbf{x},\mathbf{Z}}$ defined as $\mathbf{v} \to \langle \phi_{\mathbf{x}}, \Phi_{\mathbf{Z}}(\mathbf{v}) \rangle$ can be seen as a vector of $\mathbb{R}^M$, considering the image of any orthonormal basis of $\mathbb{R}^M$. In fact, let $\mathbf{e}_1, \ldots, \mathbf{e}_M$ be the usual basis of $\mathbb{R}^M$:

$$K_{\mathbf{x},\mathbf{Z}} \cong \begin{pmatrix} \langle \phi_{\mathbf{x}}, \Phi_{\mathbf{Z}}(\mathbf{e}_1) \rangle \\ \langle \phi_{\mathbf{x}}, \Phi_{\mathbf{Z}}(\mathbf{e}_2) \rangle \\ \vdots \\ \langle \phi_{\mathbf{x}}, \Phi_{\mathbf{Z}}(\mathbf{e}_M) \rangle \end{pmatrix} = \begin{pmatrix} \langle \phi_{\mathbf{x}}, \sum_{i=1}^M \mathbf{e}_{1,i}\phi_{\mathbf{z}_1} \rangle \\ \langle \phi_{\mathbf{x}}, \sum_{i=1}^M \mathbf{e}_{2,i}\phi_{\mathbf{z}_2} \rangle \\ \vdots \\ \langle \phi_{\mathbf{x}}, \sum_{i=1}^M \mathbf{e}_{M,i}\phi_{\mathbf{z}_M} \rangle \end{pmatrix} = \begin{pmatrix} \langle \phi_{\mathbf{x}}, \phi_{\mathbf{z}_1} \rangle \\ \langle \phi_{\mathbf{x}}, \phi_{\mathbf{z}_2} \rangle \\ \vdots \\ \langle \phi_{\mathbf{x}}, \phi_{\mathbf{z}_M} \rangle \end{pmatrix} \in \mathbb{R}^M. \tag{8}$$

Assume a variational distribution $q(\mathbf{u}) = \mathcal{N}(\mathbf{u}|\tilde{\boldsymbol{m}}, \tilde{\boldsymbol{S}})$ with $\tilde{\boldsymbol{m}} \in \mathbb{R}^M$ and $\tilde{\boldsymbol{S}} \in \mathbb{R}^{M \times M}$. Then, defining the correspondent dual vectors as

$$\boldsymbol{a} = K_{\mathbf{Z}}^{-1}\tilde{\boldsymbol{m}} \in \mathbb{R}^M, \quad \boldsymbol{A} = K_{\mathbf{Z}}^{-1}\tilde{\boldsymbol{S}}K_{\mathbf{Z}}^{-1} - K_{\mathbf{Z}}^{-1} \in \mathbb{R}^{M \times M}, \tag{9}$$

the mean and covariance functions in the dual formulation of the sparse GP $q(f)$ are

$$m^\star(\mathbf{x}) = \langle \phi_{\mathbf{x}}, \tilde{\mu} \rangle = \langle \phi_{\mathbf{x}}, \Phi_{\mathbf{Z}}(K_{\mathbf{Z}}^{-1}\tilde{\boldsymbol{m}}) \rangle = K_{\mathbf{x},\mathbf{Z}}K_{\mathbf{Z}}^{-1}\tilde{\boldsymbol{m}}, \tag{10}$$

and

$$\begin{aligned} K^\star(\mathbf{x}, \mathbf{x}') &= \langle \phi_{\mathbf{x}}, \Sigma(\phi_{\mathbf{x}'}) \rangle = \langle \phi_{\mathbf{x}}, \phi_{\mathbf{x}'} + \Phi_{\mathbf{Z}}\boldsymbol{A}\Phi_{\mathbf{Z}}^T\phi_{\mathbf{x}'} \rangle = \langle \phi_{\mathbf{x}}, \phi_{\mathbf{x}'} \rangle + \langle \phi_{\mathbf{x}}, \Phi_{\mathbf{Z}}\boldsymbol{A}\Phi_{\mathbf{Z}}^T\phi_{\mathbf{x}'} \rangle \\ &= K(\mathbf{x}, \mathbf{x}') + K_{\mathbf{x},\mathbf{Z}}\boldsymbol{A}K_{\mathbf{x}',\mathbf{Z}}^T = K(\mathbf{x}, \mathbf{x}') + K_{\mathbf{x},\mathbf{Z}}(K_{\mathbf{Z}}^{-1}\tilde{\boldsymbol{S}}K_{\mathbf{Z}}^{-1} - K_{\mathbf{Z}}^{-1})K_{\mathbf{Z},\mathbf{x}'}, \end{aligned} \tag{11}$$

which is the same approximate GP posterior $p(\mathbf{f}) = \int p(\mathbf{f}|\mathbf{u})q(\mathbf{u}) \, d\mathbf{u}$ found in Equation (6) of [Titsias (2009)](#). ∎

**Proposition 2** *If $g(\cdot, \hat{\boldsymbol{\theta}}) \in \mathcal{H}$, then $\forall \epsilon > 0$ exists a set of $M_\alpha$ inducing points $\mathbf{Z}_\alpha$ and a collection of scalar values $\boldsymbol{a} \in \mathbb{R}^{M_\alpha}$ such that the dual representation of the sparse Gaussian process defined by*

$$\tilde{\mu} = \Phi_{\mathbf{Z}_\alpha}(\boldsymbol{a}) \quad and \quad \tilde{\Sigma} = (I + \Phi_{\mathbf{Z}_\beta}\boldsymbol{A}\Phi_{\mathbf{Z}_\beta}^T)^{-1}, \tag{12}$$

corresponds to a GP posterior approximation with mean and covariance functions defined as

$$m^\star(\mathbf{x}) = h_\epsilon(\mathbf{x}) \,, \tag{13}$$
$$K^\star(\mathbf{x}, \mathbf{x}') = K(\mathbf{x}, \mathbf{x}') - K_{\mathbf{x}, \mathbf{Z}_\beta}(\boldsymbol{A}^{-1} + \boldsymbol{K}_\beta)^{-1} K_{\mathbf{Z}_\beta, \mathbf{x}'} \,,$$

where $\mathbf{Z}_\beta$ is a set of $M_\beta$ inducing points, $\boldsymbol{A} \in \mathbb{R}^{M_\beta \times M_\beta}$, $K_{\mathbf{x}, \mathbf{Z}_\beta}$ is a vector with the covariances between $f(\mathbf{x})$ and $f(\mathbf{Z}_\beta)$, and $h_\epsilon$ verifies $d_\mathcal{H}(g(\cdot, \hat{\boldsymbol{\theta}}), h_\epsilon) \leq \epsilon$, with $d_\mathcal{H}(\cdot, \cdot)$ the distance in the RKHS.

**Proof** First of all, if $g(\cdot, \hat{\boldsymbol{\theta}}) \in \mathcal{H}$, the reproducing property of the RKHS verifies that $\forall \epsilon > 0$ there exists $\mathbf{Z}_\alpha \subset \mathcal{X}$, with $\mathcal{X}$ the input space, and $\{\boldsymbol{a}_i\}_{i \in \mathbb{N}}$ such that $h_\epsilon := \sum_{i=1}^{M_\alpha} a_i \phi_{\mathbf{z}_i} = \Phi_{\mathbf{Z}_\alpha}(\boldsymbol{a})$ verifies

$$d_\mathcal{H}(g(\cdot, \hat{\boldsymbol{\theta}}), h_\epsilon) \leq \epsilon \,. \tag{14}$$

As a result, the mean function of the approximate posterior is

$$m^\star(\mathbf{x}) = \langle \phi_\mathbf{x}, \tilde{\mu} \rangle = \tilde{\mu}(\mathbf{x}) = h_\epsilon(\mathbf{x}) \approx g(\mathbf{x}, \hat{\boldsymbol{\theta}}) \,. \tag{15}$$

On the other hand, using that

$$(I + \Phi_{\mathbf{Z}_\beta} \boldsymbol{A} \Phi_{\mathbf{Z}_\beta}^T)^{-1} = I - \Phi_{\mathbf{Z}_\beta}(\boldsymbol{A}^{-1} + \Phi_{\mathbf{Z}_\beta}^T \Phi_{\mathbf{Z}_\beta})^{-1} \Phi_{\mathbf{Z}_\beta}^T \,, \tag{16}$$

the covariance function is

$$
\begin{aligned}
K^\star(\mathbf{x}, \mathbf{x}') &= \langle \phi_\mathbf{x}, \tilde{\Sigma}(\phi_{\mathbf{x}'}) \rangle = \langle \phi_\mathbf{x}, \phi_{\mathbf{x}'} \rangle - \langle \phi_\mathbf{x}, \Phi_{\mathbf{Z}_\beta}(\boldsymbol{A}^{-1} + \Phi_{\mathbf{Z}_\beta}^T \Phi_{\mathbf{Z}_\beta})^{-1} \Phi_{\mathbf{Z}_\beta}^T \phi_{\mathbf{x}'} \rangle \\
&= K(\mathbf{x}, \mathbf{x}') - K_{\mathbf{x}, \mathbf{Z}_\beta}(\boldsymbol{A}^{-1} - K_{\mathbf{Z}_\beta})^{-1} K_{\mathbf{Z}_\beta, \mathbf{x}'} \,.
\end{aligned} \tag{17}
$$

Where the characterization of $K_{\mathbf{Z}_\beta}$ and $K_{\mathbf{x}, \mathbf{Z}_\beta}$ as elements of $\mathbb{R}^{M_\beta \times M_\beta}$ and $\mathbb{R}^{M_\beta}$ of Equations (7) and (8) are used. ∎

**Proposition 3** *The value of $\boldsymbol{A}$ in Proposition 2 that minimizes Equation (5) is*

$$\boldsymbol{A} = \frac{1}{\sigma^2} \boldsymbol{K}_\beta^{-1} \boldsymbol{K}_{\mathbf{Z}_\beta, \mathbf{X}} \boldsymbol{K}_{\mathbf{X}, \mathbf{Z}_\beta} \boldsymbol{K}_\beta^{-1} \,, \tag{18}$$

*where $\sigma^2$ is the noise variance and $\boldsymbol{K}_{\mathbf{X}, \mathbf{Z}_\beta}$ is a matrix with the prior covariances between $f(\mathbf{X})$ and $f(\mathbf{Z}_\beta)$. If $\mathbf{Z}_\beta = \mathbf{X}$, the covariance function of the predictive distribution in (6) is equal to that of the full GP.*

**Proof** For simplicity, assume a non-inverse reparameterization where

$$q(f) = \mathcal{N}\left( f \,\middle|\, \Phi_{\mathbf{Z}_\alpha}(\boldsymbol{a}), I + \Phi_{\mathbf{Z}_\beta} \hat{\boldsymbol{A}} \Phi_{\mathbf{Z}_\beta} \right) \,. \tag{19}$$

We will find the optimal value for $\hat{\boldsymbol{A}}$ and compute the corresponding value for $\boldsymbol{A}$ (using the inverse reparameterization). First, we will show that the true GP posterior can be written in the dual formulation as the following

$$p(f|y) = \mathcal{N}\left( f | (\Phi_\mathbf{X} \Phi_\mathbf{X}^T + \sigma^2 \boldsymbol{I})^{-1} \Phi_\mathbf{X} y), \sigma^2 (\Phi_\mathbf{X} \Phi_\mathbf{X}^T + \sigma^2 \boldsymbol{I})^{-1} \right) \,, \tag{20}$$

where it verifies that the GP posterior mean function is

$$
\begin{aligned}
m(\mathbf{x}) &= \langle \phi_{\mathbf{x}}, (\Phi_{\boldsymbol{X}}\Phi_{\boldsymbol{X}}^T + \sigma^2 \boldsymbol{I})^{-1}\Phi_{\boldsymbol{X}} y \rangle = (\Phi_{\boldsymbol{X}}\Phi_{\boldsymbol{X}}^T + \sigma^2 \boldsymbol{I})^{-1}\Phi_{\boldsymbol{X}}\phi_{\mathbf{x}} y \\
&= (K_{\mathbf{X},\mathbf{X}} + \sigma^2 \boldsymbol{I})^{-1}K_{\mathbf{X},\mathbf{x}} y = K_{\mathbf{x},\mathbf{X}}(K_{\mathbf{X},\mathbf{X}} + \sigma^2 \boldsymbol{I})^{-1} y\,.
\end{aligned}
\tag{21}
$$

Where the characterization of $K_{\mathbf{X},\mathbf{X}}$ and $K_{\mathbf{X},\mathbf{x}}$ as elements of $\mathbb{R}^{N\times N}$ and $\mathbb{R}^N$ is used, similarly to Equations (7) and (8). Furthermore, using Woodbury matrix identity,

$$
\sigma^2 (\Phi_{\boldsymbol{X}}\Phi_{\boldsymbol{X}}^T + \sigma^2 \boldsymbol{I})^{-1} = \boldsymbol{I} - \Phi_{\boldsymbol{X}}(K_{\mathbf{X},\mathbf{X}} + \sigma^2 \boldsymbol{I})^{-1}\Phi_{\boldsymbol{X}}^T\,,
\tag{22}
$$

where again, correspondence between operators and matrices is used. This leads to the covariance function:

$$
\begin{aligned}
K(\mathbf{x}, \mathbf{x}') &= \langle \phi_{\mathbf{x}}, \sigma^2 (\Phi_{\boldsymbol{X}}\Phi_{\boldsymbol{X}}^T + \sigma^2 \boldsymbol{I})^{-1}(\phi_{\mathbf{x}'}) \rangle = \langle \phi_{\mathbf{x}}, \boldsymbol{I} - \Phi_{\boldsymbol{X}}(K_{\mathbf{X},\mathbf{X}} + \sigma^2 \boldsymbol{I})^{-1}\Phi_{\boldsymbol{X}}^T(\phi_{\mathbf{x}'}) \rangle \\
&= K_{\mathbf{x},\mathbf{x}'} - K_{\mathbf{x},\mathbf{X}}(K_{\mathbf{X},\mathbf{X}} + \sigma^2 \boldsymbol{I})^{-1}K_{\mathbf{X},\mathbf{x}'}\,.
\end{aligned}
\tag{23}
$$

This mean and covariance function are exactly the ones obtained from the original GP formulation (Titsias, 2009). Using the ELBO is equivalent to the KL divergence between the true posterior and the variational approximation, we got

$$
\mathrm{KL}(q(f)\mid p(f|y)) \propto \frac{1}{2}\mathrm{tr}\Big(\boldsymbol{B}^{-1}(\boldsymbol{I} + \Phi_{\boldsymbol{Z}_\beta}\hat{\boldsymbol{A}}\Phi_{\boldsymbol{Z}_\beta}^T)\Big) - \frac{1}{2}\ln\left(\left|\boldsymbol{I} + \Phi_{\boldsymbol{Z}_\beta}\hat{\boldsymbol{A}}\Phi_{\boldsymbol{Z}_\beta}^T\right|\right),
\tag{24}
$$

where

$$
\boldsymbol{B} = \sigma^2 (\Phi_{\boldsymbol{X}}\Phi_{\boldsymbol{X}}^T + \sigma^2 \boldsymbol{I})^{-1}\,.
\tag{25}
$$

Naming $\boldsymbol{M} = \boldsymbol{I} + \Phi_{\boldsymbol{Z}_\beta}\hat{\boldsymbol{A}}\Phi_{\boldsymbol{Z}_\beta}^T$, it is important to notice that given $\Phi_{\boldsymbol{Z}_\beta}\hat{\boldsymbol{A}}\Phi_{\boldsymbol{Z}_\beta}^T = \sum_{i=1}^M \sum_{j=1}^M \phi_{\mathbf{z}_i}\hat{a}_{i,j}\phi_{\mathbf{z}_j}^T$, despite being an operator, $\boldsymbol{M}$ can be seen as a matrix whose entries are the application of $\boldsymbol{I} + \Phi_{\boldsymbol{Z}_\beta}(\cdot)\Phi_{\boldsymbol{Z}_\beta}^T$ to the usual basis of matrices. In short

$$
\boldsymbol{M} = \boldsymbol{I} + \begin{pmatrix} \phi_{\boldsymbol{z}_1}\hat{a}_{1,1}\phi_{\boldsymbol{z}_1}^T & \cdots & \phi_{\boldsymbol{z}_1}\hat{a}_{1,M}\phi_{\boldsymbol{z}_M}^T \\ \vdots & & \vdots \\ \phi_{\boldsymbol{z}_M}\hat{a}_{M,1}\phi_{\boldsymbol{z}_1}^T & \cdots & \phi_{\boldsymbol{z}_M}\hat{a}_{M,M}\phi_{\boldsymbol{z}_M}^T \end{pmatrix} \in \mathbb{R}^{M_\beta \times M_\beta}\,.
\tag{26}
$$

The partial derivative of $\boldsymbol{M}$ w.r.t. a single position in the matrix $\hat{\boldsymbol{A}}$ is $\frac{\partial \boldsymbol{M}}{\partial \hat{a}_{ij}} = (\Phi_{\boldsymbol{Z}_\beta}\delta_i)(\Phi_{\boldsymbol{Z}_\beta}\delta_j)^T$, where $\delta_j$ denotes a zero-vector with a single 1 at position $j$. Then, using the chain rule for matrices,

$$
\frac{\partial g(U)}{\partial X_{ij}} = \mathrm{tr}\left(\frac{\partial g(U)}{\partial U}^T \frac{\partial U}{\partial X_{ij}}\right),
\tag{27}
$$

we can compute the optimum in the two terms in the KL. The optimum for the logarithm term can be computed as

$$
\begin{aligned}
\frac{\partial \ln(|\boldsymbol{M}|)}{\partial \hat{a}_{ij}} &= \mathrm{tr}\left(\frac{\partial \ln(|\boldsymbol{M}|)}{\partial \boldsymbol{M}}^T \frac{\partial \boldsymbol{M}}{\partial \hat{a}_{ij}}\right) \\
&= \mathrm{tr}(\boldsymbol{M}^{-1^T}(\Phi_{\boldsymbol{Z}_\beta}\delta_i)(\Phi_{\boldsymbol{Z}_\beta}\delta_j)^T) \\
&= \mathrm{tr}((\Phi_{\boldsymbol{Z}_\beta}\delta_j)^T \boldsymbol{M}^{-1}(\Phi_{\boldsymbol{Z}_\beta}\delta_i)) \\
&= (\Phi_{\boldsymbol{Z}_\beta}^T \boldsymbol{M}^{-1}\Phi_{\boldsymbol{Z}_\beta})\delta_{ji}\,,
\end{aligned}
\tag{28}
$$

leading to

$$\frac{\partial \ln(|\boldsymbol{M}|)}{\partial \hat{\boldsymbol{A}}} = \Phi_{\boldsymbol{Z}_\beta}^T \boldsymbol{M}^{-1} \Phi_{\boldsymbol{Z}_\beta} = \Phi_{\boldsymbol{Z}_\beta}^T (\boldsymbol{I} + \Phi_{\boldsymbol{Z}_\beta} \hat{\boldsymbol{A}} \Phi_{\boldsymbol{Z}_\beta}^T)^{-1} \Phi_{\boldsymbol{Z}_\beta} \,. \tag{29}$$

On the other hand, the optimum for the trace term is

$$\begin{aligned}
\frac{\partial \mathrm{tr}(\boldsymbol{B}^{-1}\boldsymbol{M})}{\partial \hat{a}_{ij}} &= \mathrm{tr}\left(\frac{\partial \mathrm{tr}(\boldsymbol{B}^{-1}\boldsymbol{M})}{\partial \boldsymbol{M}}^T \frac{\partial \boldsymbol{M}}{\partial \hat{a}_{ij}}\right) \\
&= \mathrm{tr}\left(\boldsymbol{B}^{-1}(\Phi_{\boldsymbol{Z}_\beta}\delta_i)(\Phi_{\boldsymbol{Z}_\beta}\delta_j)^T\right) \\
&= \mathrm{tr}((\Phi_{\boldsymbol{Z}_\beta}\delta_j)^T \boldsymbol{B}^{-1^T}(\Phi_{\boldsymbol{Z}_\beta}\delta_i)) \\
&= (\Phi_{\boldsymbol{Z}_\beta}^T \boldsymbol{B}^{-1} \Phi_{\boldsymbol{Z}_\beta})\delta_{ji} \,,
\end{aligned} \tag{30}$$

where we used that $\boldsymbol{B} = \boldsymbol{B}^T$. As a result,

$$\begin{aligned}
\frac{\partial \mathrm{tr}(\boldsymbol{B}^{-1}\boldsymbol{M})}{\partial \hat{\boldsymbol{A}}} &= \Phi_{\boldsymbol{Z}_\beta}^T \boldsymbol{B}^{-1} \Phi_{\boldsymbol{Z}_\beta} \\
&= \sigma^2 \Phi_{\boldsymbol{Z}_\beta}^T (\Phi_{\boldsymbol{X}}\Phi_{\boldsymbol{X}}^T + \sigma^2\boldsymbol{I})^{-1} \Phi_{\boldsymbol{Z}_\beta} \,.
\end{aligned} \tag{31}$$

Using all the derivations

$$\frac{\partial \mathrm{KL}(q(f) \mid p(f|y))}{\partial \hat{\boldsymbol{A}}} = 0 \iff \Phi_{\boldsymbol{Z}_\beta}^T (\boldsymbol{B}^{-1} - (\boldsymbol{I} + \Phi_{\boldsymbol{Z}_\beta} \hat{\boldsymbol{A}} \Phi_{\boldsymbol{Z}_\beta}^T)^{-1})\Phi_{\boldsymbol{Z}_\beta} = 0 \tag{32}$$

Using the Woodbury matrix identity

$$\begin{aligned}
0 &= \Phi_{\boldsymbol{Z}_\beta}^T (\boldsymbol{B}^{-1} - (\boldsymbol{I} + \Phi_{\boldsymbol{Z}_\beta} \hat{\boldsymbol{A}} \Phi_{\boldsymbol{Z}_\beta}^T)^{-1})\Phi_{\boldsymbol{Z}_\beta} \\
&= \Phi_{\boldsymbol{Z}_\beta}^T (\boldsymbol{B}^{-1} - \boldsymbol{I} + \Phi_{\boldsymbol{Z}_\beta}(\hat{\boldsymbol{A}}^{-1} + \boldsymbol{K}_\beta)^{-1}\Phi_{\boldsymbol{Z}_\beta}^T)\Phi_{\boldsymbol{Z}_\beta} \\
&= \Phi_{\boldsymbol{Z}_\beta}^T \boldsymbol{B}^{-1}\Phi_{\boldsymbol{Z}_\beta} - \boldsymbol{K}_\beta + \boldsymbol{K}_\beta(\hat{\boldsymbol{A}}^{-1} + \boldsymbol{K}_\beta)^{-1}\boldsymbol{K}_\beta \,.
\end{aligned} \tag{33}$$

Thus, the value of $\hat{\boldsymbol{A}}$ where $\partial \mathrm{KL}/\partial \hat{\boldsymbol{A}} = 0$ verifies

$$\hat{\boldsymbol{A}} = ((\boldsymbol{K}_\beta^{-1} - \boldsymbol{K}_\beta^{-1}\Phi_{\boldsymbol{Z}_\beta}^T \boldsymbol{B}^{-1}\Phi_{\boldsymbol{Z}_\beta}\boldsymbol{K}_\beta^{-1})^{-1} - \boldsymbol{K}_\beta)^{-1} \,. \tag{34}$$

Using that $\boldsymbol{B}^{-1} = \sigma^{-2}(\Phi_{\boldsymbol{X}}\Phi_{\boldsymbol{X}}^T + \sigma^2\boldsymbol{I})$, we can take further derivations on the expression of $\hat{\boldsymbol{A}}$ as

$$\begin{aligned}
\hat{\boldsymbol{A}} &= ((\boldsymbol{K}_\beta^{-1} - \boldsymbol{K}_\beta^{-1}\Phi_{\boldsymbol{Z}_\beta}^T \boldsymbol{B}^{-1}\Phi_{\boldsymbol{Z}_\beta}\boldsymbol{K}_\beta^{-1})^{-1} - \boldsymbol{K}_\beta)^{-1} \\
&= ((\boldsymbol{K}_\beta^{-1} - \boldsymbol{K}_\beta^{-1}\Phi_{\boldsymbol{Z}_\beta}^T \sigma^{-2}(\Phi_{\boldsymbol{X}}\Phi_{\boldsymbol{X}}^T + \sigma^2\boldsymbol{I})\Phi_{\boldsymbol{Z}_\beta}\boldsymbol{K}_\beta^{-1})^{-1} - \boldsymbol{K}_\beta)^{-1} \\
&= ((\boldsymbol{K}_\beta^{-1} - \sigma^{-2}\boldsymbol{K}_\beta^{-1}\boldsymbol{K}_{\boldsymbol{Z}_\beta,\boldsymbol{X}}\boldsymbol{K}_{\boldsymbol{X},\boldsymbol{Z}_\beta}\boldsymbol{K}_\beta^{-1} - \boldsymbol{K}_\beta^{-1})^{-1} - \boldsymbol{K}_\beta)^{-1} \\
&= ((-\sigma^{-2}\boldsymbol{K}_\beta^{-1}\boldsymbol{K}_{\boldsymbol{Z}_\beta,\boldsymbol{X}}\boldsymbol{K}_{\boldsymbol{X},\boldsymbol{Z}_\beta}\boldsymbol{K}_\beta^{-1})^{-1} - \boldsymbol{K}_\beta)^{-1} \\
&= -((\sigma^{-2}\boldsymbol{K}_\beta^{-1}\boldsymbol{K}_{\boldsymbol{Z}_\beta,\boldsymbol{X}}\boldsymbol{K}_{\boldsymbol{X},\boldsymbol{Z}_\beta}\boldsymbol{K}_\beta^{-1})^{-1} + \boldsymbol{K}_\beta)^{-1} \,.
\end{aligned} \tag{35}$$

Applying again Woodbury matrix identity:

$$\begin{aligned}
\hat{\boldsymbol{A}} &= -((\sigma^{-2}\boldsymbol{K}_\beta^{-1}\boldsymbol{K}_{\boldsymbol{Z}_\beta,\boldsymbol{X}}\boldsymbol{K}_{\boldsymbol{X},\boldsymbol{Z}_\beta}\boldsymbol{K}_\beta^{-1})^{-1} + \boldsymbol{K}_\beta)^{-1} \\
&= -(\boldsymbol{K}_\beta^{-1} - \boldsymbol{K}_\beta^{-1}(\sigma^{-2}\boldsymbol{K}_\beta^{-1}\boldsymbol{K}_{\boldsymbol{Z}_\beta,\boldsymbol{X}}\boldsymbol{K}_{\boldsymbol{X},\boldsymbol{Z}_\beta}\boldsymbol{K}_\beta^{-1} + \boldsymbol{K}_\beta^{-1})^{-1}\boldsymbol{K}_\beta^{-1}) \\
&= -(\boldsymbol{K}_\beta^{-1} - (\sigma^{-2}\boldsymbol{K}_{\boldsymbol{Z}_\beta,\boldsymbol{X}}\boldsymbol{K}_{\boldsymbol{X},\boldsymbol{Z}_\beta} + \boldsymbol{K}_\beta)^{-1}) \\
&= -\boldsymbol{K}_\beta^{-1} + (\sigma^{-2}\boldsymbol{K}_{\boldsymbol{Z}_\beta,\boldsymbol{X}}\boldsymbol{K}_{\boldsymbol{X},\boldsymbol{Z}_\beta} + \boldsymbol{K}_\beta)^{-1} \,.
\end{aligned} \tag{36}$$

If we substitute this value on the predictive distribution

$$
\begin{aligned}
K^\star(\mathbf{x}, \mathbf{x}') = \langle \phi_\mathbf{x}, \boldsymbol{M}(\phi_{\mathbf{x}'}) \rangle &= K(\mathbf{x}, \mathbf{x}') + K_{\mathbf{x}, \boldsymbol{Z}_\beta} \hat{\boldsymbol{A}} K_{\boldsymbol{Z}_\beta, \mathbf{x}'} \\
&= K(\mathbf{x}, \mathbf{x}') + K_{\mathbf{x}, \mathbf{Z}}(-\boldsymbol{K}_\beta^{-1} + (\sigma^{-2} \boldsymbol{K}_{\boldsymbol{Z}_\beta, \boldsymbol{X}} \boldsymbol{K}_{\boldsymbol{X}, \boldsymbol{Z}_\beta} + \boldsymbol{K}_\beta)^{-1}) K_{\boldsymbol{Z}_\beta, \mathbf{x}'} \\
&= K(\mathbf{x}, \mathbf{x}') - K_{\mathbf{x}, \boldsymbol{Z}_\beta} \boldsymbol{K}_\beta^{-1} K_{\boldsymbol{Z}_\beta, \mathbf{x}'} + K_{\mathbf{x}, \boldsymbol{Z}_\beta} (\sigma^{-2} \boldsymbol{K}_{\boldsymbol{Z}_\beta, \boldsymbol{X}} \boldsymbol{K}_{\boldsymbol{X}, \boldsymbol{Z}_\beta} + \boldsymbol{K}_\beta)^{-1}) K_{\boldsymbol{Z}_\beta, \mathbf{x}'} \\
&= K(\mathbf{x}, \mathbf{x}') - K_{\mathbf{x}, \boldsymbol{Z}_\beta} \boldsymbol{K}_\beta^{-1} K_{\boldsymbol{Z}_\beta, \mathbf{x}'} \\
&\quad + K_{\mathbf{x}, \boldsymbol{Z}_\beta} \boldsymbol{K}_\beta^{-1} (\boldsymbol{K}_\beta (\sigma^{-2} \boldsymbol{K}_{\boldsymbol{Z}_\beta, \boldsymbol{X}} \boldsymbol{K}_{\boldsymbol{X}, \boldsymbol{Z}_\beta} + \boldsymbol{K}_\beta)^{-1} \boldsymbol{K}_\beta) \boldsymbol{K}_\beta^{-1} K_{\boldsymbol{Z}_\beta, \mathbf{x}'} \,.
\end{aligned}
\tag{37}
$$

This expression coincides with the optimal sparse GP solution described by Titsias (2009), with optimal variational covariance $(\boldsymbol{K}_\beta (\sigma^{-2} \boldsymbol{K}_{\boldsymbol{Z}_\beta, \boldsymbol{X}} \boldsymbol{K}_{\boldsymbol{X}, \boldsymbol{Z}_\beta} + \boldsymbol{K}_\beta)^{-1} \boldsymbol{K}_\beta)$. As a result, the optimal solution for VaLLA coincides with the optimal solution for standard sparse GPs. Let us now compute the optimal value of $\boldsymbol{A}$ given the optimal value of $\hat{\boldsymbol{A}}$. First, notice that using Woodbury Matrix identity

$$
(\boldsymbol{I} + \Phi_{\boldsymbol{Z}_\beta} \boldsymbol{A} \Phi_{\boldsymbol{Z}_\beta}^T)^{-1} = \boldsymbol{I} - \Phi_{\boldsymbol{Z}_\beta} (\boldsymbol{A} + \boldsymbol{K}_\beta)^{-1} \Phi_{\boldsymbol{Z}_\beta}^T \,.
\tag{38}
$$

Therefore, the relation between $\boldsymbol{A}$ and $\hat{\boldsymbol{A}}$ is $\hat{\boldsymbol{A}} = -(\boldsymbol{A} + \boldsymbol{K}_\beta)^{-1}$. Meaning that

$$
\hat{\boldsymbol{A}} = -\boldsymbol{K}_\beta^{-1} + (\sigma^{-2} \boldsymbol{K}_{\boldsymbol{Z}_\beta, \boldsymbol{X}} \boldsymbol{K}_{\boldsymbol{X}, \boldsymbol{Z}_\beta} + \boldsymbol{K}_\beta)^{-1} \implies \boldsymbol{A} = \frac{1}{\sigma^2} \boldsymbol{K}_\beta^{-1} \boldsymbol{K}_{\boldsymbol{Z}_\beta, \boldsymbol{X}} \boldsymbol{K}_{\boldsymbol{X}, \boldsymbol{Z}_\beta} \boldsymbol{K}_\beta^{-1} \,.
\tag{39}
$$

**Global optimum** To complete the proof of the solution in (39) being not only optimal but also a maximum of the ELBO, we must test the behavior of the second derivative w.r.t. $\boldsymbol{A}$. Let us reuse the previous results, where we found that

$$
\frac{\partial \mathrm{KL}(q(f) \mid p(f|y))}{\partial \hat{a}_{i,j}} = \frac{1}{2} \delta_j^T (\Phi_{\boldsymbol{Z}_\beta}^T \boldsymbol{B}^{-1} \Phi_{\boldsymbol{Z}_\beta} - \Phi_{\boldsymbol{Z}_\beta}^T \boldsymbol{M}^{-1} \Phi_{\boldsymbol{Z}_\beta}) \delta_i
\tag{40}
$$

Taking a second derivative w.r.t. another location $\hat{a}_{u,v}$ yields

$$
\frac{\partial}{\partial \hat{a}_{u,v}} \frac{\partial \mathrm{KL}(q(f) \mid p(f|y))}{\partial \hat{a}_{i,j}} = \frac{1}{2} \frac{\partial}{\partial \hat{a}_{u,v}} \delta_j^T (\Phi_{\boldsymbol{Z}_\beta}^T \boldsymbol{B}^{-1} \Phi_{\boldsymbol{Z}_\beta} - \Phi_{\boldsymbol{Z}_\beta}^T \boldsymbol{M}^{-1} \Phi_{\boldsymbol{Z}_\beta}) \delta_i
\tag{41}
$$

Considering that $\boldsymbol{B}$ does not depend on $\hat{\boldsymbol{A}}$, the first term drops from the derivative. Thus

$$
\frac{\partial}{\partial \hat{a}_{u,v}} \frac{\partial \mathrm{KL}(q(f) \mid p(f|y))}{\partial \hat{a}_{i,j}} = -\frac{1}{2} \frac{\partial}{\partial \hat{a}_{u,v}} \delta_j^T (\Phi_{\boldsymbol{Z}_\beta}^T \boldsymbol{M}^{-1} \Phi_{\boldsymbol{Z}_\beta}) \delta_i
\tag{42}
$$

Here we aim to use the chain rule for matrices,

$$
\frac{\partial g(U)}{\partial X_{ij}} = \mathrm{tr}\left( \frac{\partial g(U)}{\partial U}^T \frac{\partial U}{\partial X_{ij}} \right),
\tag{43}
$$

Then, consider that

$$
\frac{\partial \, \delta_j^T (\Phi_{\boldsymbol{Z}_\beta}^T \boldsymbol{M}^{-1} \Phi_{\boldsymbol{Z}_\beta}) \delta_i}{\partial \boldsymbol{M}} = -\boldsymbol{M}^{-1} \Phi_{\boldsymbol{Z}_\beta} \delta_j \delta_i^T \Phi_{\boldsymbol{Z}_\beta}^T \boldsymbol{M}^{-1}
\tag{44}
$$

Then,

$$\frac{\partial}{\partial \hat{a}_{u,v}} \delta_j^T (\Phi_{\boldsymbol{Z}_\beta}^T \boldsymbol{M}^{-1} \Phi_{\boldsymbol{Z}_\beta}) \delta_i = \text{tr}\left( \left( -\boldsymbol{M}^{-1} \Phi_{\boldsymbol{Z}_\beta} \delta_j \delta_i^T \Phi_{\boldsymbol{Z}_\beta}^T \boldsymbol{M}^{-1} \right)^T \Phi_{\boldsymbol{Z}_\beta} \delta_u (\Phi_{\boldsymbol{Z}_\beta} \delta_v)^T \right) \quad (45)$$

We can work on this trace to simplify the expression as

$$
\begin{aligned}
\frac{\partial}{\partial \hat{a}_{u,v}} \delta_j^T (\Phi_{\boldsymbol{Z}_\beta}^T \boldsymbol{M}^{-1} \Phi_{\boldsymbol{Z}_\beta}) \delta_i &= \text{tr}\left( \left( -\boldsymbol{M}^{-1} \Phi_{\boldsymbol{Z}_\beta} \delta_j \delta_i^T \Phi_{\boldsymbol{Z}_\beta}^T \boldsymbol{M}^{-1} \right)^T \Phi_{\boldsymbol{Z}_\beta} \delta_u (\Phi_{\boldsymbol{Z}_\beta} \delta_v)^T \right) \\
&= -\text{tr}\left( \delta_v^T \Phi_{\boldsymbol{Z}_\beta}^T \left( \boldsymbol{M}^{-1} \Phi_{\boldsymbol{Z}_\beta} \delta_j \delta_i^T \Phi_{\boldsymbol{Z}_\beta}^T \boldsymbol{M}^{-1} \right)^T \Phi_{\boldsymbol{Z}_\beta} \delta_u \right) \\
&= -\delta_v^T \Phi_{\boldsymbol{Z}_\beta}^T \left( \boldsymbol{M}^{-1} \Phi_{\boldsymbol{Z}_\beta} \delta_j \delta_i^T \Phi_{\boldsymbol{Z}_\beta}^T \boldsymbol{M}^{-1} \right)^T \Phi_{\boldsymbol{Z}_\beta} \delta_u \\
&= -\delta_v^T \Phi_{\boldsymbol{Z}_\beta}^T \boldsymbol{M}^{-1} \Phi_{\boldsymbol{Z}_\beta} \delta_j \delta_i^T \Phi_{\boldsymbol{Z}_\beta}^T \boldsymbol{M}^{-1} \Phi_{\boldsymbol{Z}_\beta} \delta_u
\end{aligned}
\quad (46)
$$

Naming $\boldsymbol{Q} = \Phi_{\boldsymbol{Z}_\beta}^T \boldsymbol{M}^{-1} \Phi_{\boldsymbol{Z}_\beta}$, we got

$$\frac{\partial}{\partial \hat{a}_{u,v}} \frac{\partial \text{KL}(q(f) \mid p(f|y))}{\partial \hat{a}_{i,j}} = \frac{1}{2} \boldsymbol{Q}_{v,j} \cdot \boldsymbol{Q}_{i,u} = \frac{1}{2} \boldsymbol{Q}_{j,v} \cdot \boldsymbol{Q}_{i,u} \quad (47)$$

where in the last equality we used that $\boldsymbol{Q}$ is symmetric. Using the definition of $\boldsymbol{M}$ we know that

$$\boldsymbol{M}^{-1} = \boldsymbol{I} - \Phi_{\boldsymbol{Z}_\beta} (\boldsymbol{A} + \boldsymbol{K}_\beta)^{-1} \Phi_{\boldsymbol{Z}_\beta}^T \implies \boldsymbol{Q} = \boldsymbol{K}_\beta - \boldsymbol{K}_\beta (\boldsymbol{A} + \boldsymbol{K}_\beta)^{-1} \boldsymbol{K}_\beta \quad (48)$$

This shows that $\boldsymbol{Q}$ is the posterior covariance of a GP with prior covariances $\boldsymbol{K}_\beta$ and noise covariances $\boldsymbol{A}$.

$$\frac{\partial}{\partial \hat{\boldsymbol{A}}} \frac{\partial \text{KL}(q(f) \mid p(f|y))}{\partial \hat{\boldsymbol{A}}} = \frac{1}{2} \boldsymbol{Q} \otimes \boldsymbol{Q} \quad (49)$$

with $\otimes$ denoting the Kronecker product and $\boldsymbol{Q}$ a definite positive matrix. As the Kronecker product of two definite positive matrices is definite positive, the optimal is a minimum of the KL. Given that this second derivative is positive-definite, the value found is the global minimum of the ELBO objective.

■

## Appendix B. Related Work

LA for DNNs was originally introduced by Mackay (1992), applying it to small networks using the full Hessian. MacKay (1992) also proposed an approximation similar to the generalized Gauss-Newton (GGN). The combination of scalable factorizations or diagonal Hessian approximations (Martens and Grosse, 2015; Botev et al., 2017) with the GGN approximation (Martens, 2020) played a crucial role in the resurgence of LA for modern DNNs (Ritter et al., 2018; Khan et al., 2019). Recent works aim to relax the Gaussian assumption of LLA adopting a Riemannian-Laplace approximation, where samples naturally fall into weight regions with low negative log-posterior (Bergamin et al., 2023).

To address the underfitting issue associated with LA (Lawrence, 2001), particularly when combined with the GGN approximation, Ritter et al. (2018) proposed a Kronecker factored (KFAC) LLA approximation. This approach outperforms LA with a diagonal Hessian matrix.

The GP interpretation of LLA (Khan et al., 2019) allows using GP approximate methods to speed up the computations. Immer et al. (2021) propose to use a subset of the training dataset as a scalable alternative to the true GP. Lee et al. (2022) propose a Mixture of Experts approach where each expert is trained on a different soft-margin cluster. However, the proposed clustering algorithm, although more efficient than Kernel-K-means, has linear cost w.r.t. the training set size. VaLLA, on the other hand, has sub-linear training time w.r.t. training set size due to mini-batch training. Moreover, it is not clear how to consider neighboring clusters in high dimensional input spaces. The authors only provide code for a 1-dimensional problem. Third, fitting a local GP using the data of the corresponding cluster and its neighbors is expected to overestimate the predictive variance since the model has been trained with a smaller number of training instances. This is particularly the case in datasets with millions of training instances such as Taxi. This problem is also described by Immer et al. (2021).

Deng et al. (2022) proposed a Nyström approximation of the true GP covariance matrix by using $M \ll N$ points chosen at random from the training set. The method, called ELLA, has cost $\mathcal{O}(NM^3)$. ELLA also requires computing the costly Jacobian vectors required in VaLLA, but does not need their gradients. Unlike VaLLA, the Nyström approximation needs to visit each instance in the training set. However, as stated by Deng et al. (2022), ELLA suffers from over-fitting. An early-stopping strategy, using a validation set, is proposed to alleviate it. In this case, ELLA only considers a subset of the training data. ELLA does not allow for hyper-parameter optimization, unlike VaLLA. The prior variance $\sigma_0^2$ must be tuned using grid search and a validation set, which increases training time significantly.

The recent work of Scannell et al. (2024) proposes a similar approach to VaLLA, where an inducing point sparse approach is used to construct a GP from a pre-trained DNN. However, two main points differentiate this work from our approach: (i) the pre-trained DNN is not kept as the posterior mean of the model, potentially losing prediction performance and also departing from LLA's post-hoc nature and goal; (ii) instead of using mini-batches to optimize variational parameters, they perform a full iteration over the training data to find optimal variational parameters. Thus, this results in a potentially slower method than VaLLA, which due to early-stopping and stochastic optimization, can avoid iterating over the full dataset.

Samples from a GP posterior can be efficiently computed using stochastic optimization, eluding the explicit inversion of the kernel matrix (Lin et al., 2024). This approach can be extended to LLA to generate samples from the GP posterior, avoiding the $\mathcal{O}(N^3)$ cost (Antorán et al., 2023). However, this method cannot provide an estimate of the log-marginal likelihood for hyper-parameter optimization. To address this limitation, Antorán et al. (2023) propose using the *EM-algorithm*, where samples are generated (E-step) and hyper-parameters are optimized afterwards (M-step) iteratively. The EM algorithm significantly increases computational cost, as generating a single sample is as expensive as training the original DNN on the full data. Finally, the method of Antorán et al. (2023) only considers classification problems.

## Appendix C. MAP solution in Hilbert Space

Proposition 2 assumes that $g(\cdot, \hat{\boldsymbol{\theta}}) \in \mathcal{H}$. In practice, this need not be the case. Covariance functions such as squared exponential or Matérn are recognized for encompassing the entire space of continuous functions. However, whether $g(\cdot, \hat{\boldsymbol{\theta}}) \in \mathcal{H}$ holds in general remains unknown. We assume that if $g(\cdot, \hat{\boldsymbol{\theta}}) \notin \mathcal{H}$, then $\mathcal{H}$ is sufficiently expressive to include a close approximation to $g(\cdot, \hat{\boldsymbol{\theta}})$. Consequently, $g(\cdot, \hat{\boldsymbol{\theta}})$ can be used as the sparse GP posterior mean.

Whether the map solution is in the Hilbert space might be difficult (if not impossible) to know. However, there are cases where it can be theoretically shown; for example in a linear model. Let the map solution be a linear model as

$$g(\mathbf{x}, (\boldsymbol{w}, b)) = \boldsymbol{w}^T \mathbf{x} + b \,.$$

Then, the features (Jacobians) are

$$\phi(\mathbf{x}) = (\mathbf{x}^T, 1)^T \,.$$

With a single inducing point $\mathbf{z}$ and scalar value $a \in \mathbb{R}$, the mean function would be

$$m(\mathbf{x}) = a\phi(\mathbf{x})^T \phi(\mathbf{z}) = a(\mathbf{x}^T \mathbf{z} + 1) \,.$$

This recovers the MAP solution if $a = b$ and $\mathbf{z} = \boldsymbol{w}/a$. However, if the model is not linear but has a linear last layer as

$$g(\mathbf{x}, (\boldsymbol{w}, b, \boldsymbol{\theta})) = \boldsymbol{w}^T h_{\boldsymbol{\theta}}(\mathbf{x}) + b \,,$$

where $h_{\boldsymbol{\theta}}$ is a non-linear function that depends on parameters $\boldsymbol{\theta}$. Then, the features (Jacobians) are

$$\phi(\mathbf{x}) = \left( h_{\boldsymbol{\theta}}(\mathbf{x})^T, 1, (\nabla_{\boldsymbol{\theta}} h_{\boldsymbol{\theta}}(\mathbf{x}))^T \boldsymbol{w} \right)^T \,.$$

Here it might be difficult to check if there exists a combination that yields the map solution as the mean function.

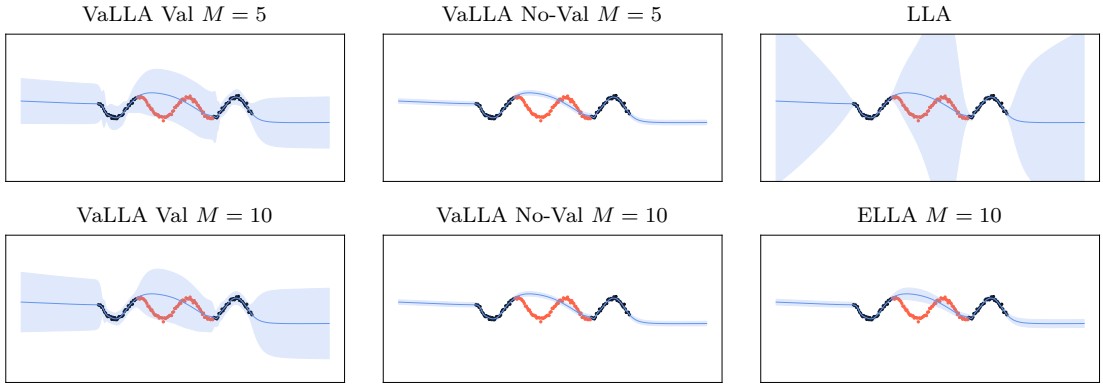

Figure 2: Predictive distribution (mean and two times the standard deviation) on a toy 1D regression dataset with a 2 hidden layer MLP with 50 units. Training points are shown in black and the validation set is shown in orange. The first column shows the obtained predictive distribution using early-stopping and a validation set, with 5 and 10 inducing points. The second column shows the results obtained without early-stopping. LLA and ELLA are shown in the last column.

## Appendix D. Over-fitting and Early Stopping

In Section 3 we mentioned the fact that the standard maximization of the ELBO leads to null predictive variances. The optimal value for the prior variance is infinite as a result of the mean being fixed to the optimal MAP solution. In a regression scenario with Gaussian noise with variance $\sigma^2$ the first term in the r.h.s. of (5) becomes:

$$\sum_{i=1}^N -\frac{\log(2\pi\sigma^2)}{2} - \frac{(y_i - g(\mathbf{x}_i, \hat{\boldsymbol{\theta}}))^2}{2\sigma^2} - \frac{K^\star(\mathbf{x}_i, \mathbf{x}_i)}{2\sigma^2} \tag{50}$$

where $(y_i - g(\mathbf{x}_i, \hat{\boldsymbol{\theta}}))^2$ is constant. Maximizing (50) w.r.t. $\sigma_0^2$ results in the prior covariances, $\sigma_0^2 J_{\hat{\boldsymbol{\theta}}}(\mathbf{x})^{\mathrm{T}} J_{\hat{\boldsymbol{\theta}}}(\mathbf{x}')$, tending to 0. This makes posterior covariances $K^\star(\mathbf{x}_i, \mathbf{x}_i)$ also tend to 0, effectively cancelling the last term in (50). The KL term in (5) is also optimal and 0 if $\sigma_0^2 \to 0$.

We circumvent this by applying $\alpha$-divergences, which are not ill-defined in this learning setup; allowing the optimization of the prior. However, the use of this optimization objective is not perfect and we faced the fact that it tends to over-fit the prior variance to the training data. The middle column of Figure 2 (middle) shows the obtained predictive distribution (two times standard deviation) learned from VaLLA using the black points as training data. The MAP solution is obtained using a 2 hidden layer MLP with 50 hidden units and *tanh* activation, optimized to minimize the RMSE of the training data for 10000 iterations with Adam and learning rate $10^{-3}$. VaLLA on the other hand is trained for 20000 iterations. As one may see in the image, the prior variance is fitted to the data to the point where the uncertainty does not increase in the middle gap of the data.

In this experiment, VaLLA optimizes hyper-parameters along with the variational objective. LLA optimizes the prior variance and likelihood variance by maximizing the marginal

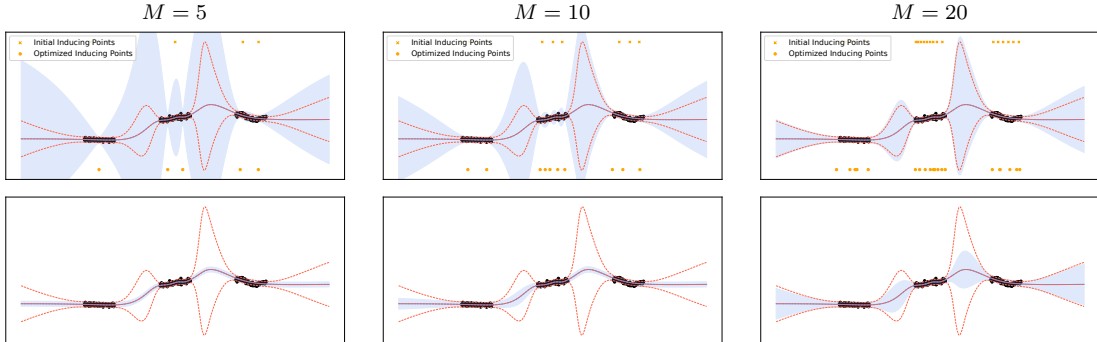

Figure 3: Predictive distribution (two times the standard deviation) on a toy 1D regression dataset with a 2 hidden layer MLP with 50 units. The obtained results for 5, 10 and 20 inducing points for VaLLA are shown in the first row. ELLA's predictive distribution with the same amount of samples from the training data is shown in the second row. LLA's predictive distribution is shown in dotted orange.

log likelihood and ELLA uses LLA's optimal hyper-parameters. In this situation there are two simple courses of action: use the ELBO and choose the prior variance by cross-validation; or perform early-stopping with the $\alpha$-divergences objective, using a validation set to stop training before over-fitting. The latter may not work as it assumes that there is a point during training where the prior variance truly explains the underlying data without over-fitting. However, as the prior variance is set to a relatively large value compared to the optimal one (which is small and leads to over-fitting), this method resulted in great performance for VaLLA. The left column of Figure 2 shows the obtained predictive distribution (two times standard deviation) learned from VaLLA, in this case, using the black points as training data and the orange points as the validation set. In the experiments, we computed the NLL of the validation set every 100 training iterations and stopped training when it worsens. This also allowed us to save computational time.

## Appendix E. Increasing Inducing Points

Using the optimal covariance in Proposition 3, if the set of inducing points equals the training points $\mathbf{Z} = \mathbf{X}$, the posterior distribution of VaLLA equals that of the exact LLA Gaussian Process. This suggest that increasing the number of inducing points would lead to better uncertainty estimations. In this section, we aim to show how close is the predictive distribution of VaLLA to that of LLA when we increase the number of inducing points.

Figure 3 shows the obtained predictive distribution of VaLLA (first row) and ELLA (second row) for $M = 5$, $M = 10$ and $M = 20$ inducing points/samples. The initial and final locations of the inducing points are also shown for VaLLA. The posterior distribution obtained by LLA is shown in dotted orange. The MAP solution is obtained using a 2 hidden layer MLP with 50 hidden units and *tanh* activation, optimized to minimize the RMSE of the training data for 12000 iterations with Adam and learning rate $10^{-3}$. VaLLA on the

**Algorithm 1:** VaLLA's training loop with $\alpha = 1$

**Require:** Pre-trained MAP solution $f$, input batch $\mathbf{X}, \mathbf{y}$, $m$ number of inducing points and $T$ iterations.

$\mathbf{Z} \leftarrow kmeans(\mathbf{X}, m)$              {Initialize inducing points}

$\boldsymbol{L} \leftarrow \boldsymbol{I}$           {Initialize Cholesky decomposition of $\boldsymbol{A}$}

**for** $i \in \{0, \dots, T-1\}$ **do**

  $(\mathbf{X}_b, \mathbf{y}_b) \leftarrow get\_batch()$           {Get mini-batch of data}

  $\boldsymbol{J_x} \leftarrow compute\_jacobian(\mathbf{X}_b)$         {Compute Jacobians}

  $\boldsymbol{J_z} \leftarrow compute\_jacobian(\mathbf{Z})$

  $\boldsymbol{K_x} \leftarrow \sigma_0^2 \boldsymbol{J_x} \boldsymbol{J_x}^T$           {Compute Kernels}

  $\boldsymbol{K_{xz}} \leftarrow \sigma_0^2 \boldsymbol{J_x} \boldsymbol{J_z}^T$

  $\boldsymbol{K_z} \leftarrow \sigma_0^2 \boldsymbol{J_z} \boldsymbol{J_z}^T$

  $\boldsymbol{A} \leftarrow \boldsymbol{L} \boldsymbol{L}^T$           {Compute Variational Matrix}

  $Q\_mean \leftarrow f(\mathbf{X}_b)$          {Compute posterior mean}

  $Q\_var \leftarrow \boldsymbol{K_x} - \boldsymbol{K_{xz}}(\boldsymbol{A}^{-1} + \boldsymbol{K_z})^{-1} \boldsymbol{K_{xz}^T}$   {Compute posterior covariance matrix}

  $KL \leftarrow compute\_KL(\boldsymbol{A}, \boldsymbol{K_z})$     {Compute Kullback-Leibler divergence}

  $NLL \leftarrow compute\_NLL(\mathbf{y}_n, Q\_mean, Q\_var)$   {Compute Negative Log-likelihood}

  $loss \leftarrow -\frac{len(\mathbf{X})}{len(\mathbf{X}_b)} NLL + KL$

  Optimize parameters by minimizing $loss$.

**end for**

other hand is trained for 30000 iterations. For this experiment, VaLLA and ELLA use the optimal prior variance and likelihood variance obtained by optimizing LLA's marginal log likelihood. As one may see in the image, it is clear that one of the main differences between the two methods is that VaLLA tends to over-estimate the variance whereas ELLA tends to infra-estimate it, compared to LLA. Furthermore, the value of $M$ for which the model is closer to the LLA posterior is lower for VaLLA than for ELLA. As we increase $M$, VaLLA's predictive distribution becomes closer and closer to that of LLA.

The initial and final position of the inducing locations is also shown in the figure. For this experiments, the initial values are computed using K-Means. It can be seen how VaLLA is capable of tuning the inducing locations and move them from one cluster of points to another as needed. This is one of the main advantages of this method compared to ELLA.

## Appendix F. Pseudocode

Algorithm 1 shows the structure of VaLLA's training loop, where no Early-Stopping is considered. Using the kernels and $\boldsymbol{A}$ it is easy to compute the KL in Eq. 5. As a result, the training loop is easy to implement, since $q(f)$, given by `Q_mean` and `Q_var` are easily computable as detailed in the algorithm.

## Appendix G. Further Experiments

We compare VaLLA with other methods using the LLA implementation by Daxberger et al. (2021a). VaLLA utilizes a batch size of 100. For regression, MNIST and FMNIST problems, we train our own DNN (standard multi-layer perceptron), which is stored for reproducibility. In the CIFAR10 experiments with ResNet, we extract the results for other methods from Deng et al. (2022) and use the same DNN for VaLLA. Hyper-parameters in all LLA variants (diagonal, KFAC, last-layer LLA) are optimized based on the marginal log-likelihood estimate. Additional experimental details are given in Appendix H.

### G.1. Synthetic Regression

We compare the predictive distribution of VaLLA with that of LLA (which is considered the optimal method), other LLA variants and ELLA, on the 1-D regression problem of Izmailov et al. (2020). In ELLA and VaLLA, we use the optimal hyper-parameters from LLA. The results in Figure 1 illustrate that VaLLA's predictive distribution closely aligns with that of LLA. Figure 3 (see Appendix E) depicts the predictive distributions of VaLLA and ELLA for varying numbers of inducing points and points in the Nyström approximation, respectively. It shows that VaLLA converges to the true posterior faster than ELLA, with VaLLA tending to over-estimate the predictive variance while ELLA under-estimates it. In Figure 2 (see Appendix E) we observe the effect of tuning the prior variance in VaLLA in another toy 1-D problem, with and without early-stopping. Notably, early stopping, using a validation set, prevents overly small predictive variances in VaLLA. Finally, we observe that when VaLLA estimates the prior variance by maximizing the $\alpha$-divergence, it tends to underestimate LLA's predictive variance.

### G.2. Airline, Year and Taxi Regression Problems

We carry out experiments on large regression datasets. (i) The *Year* dataset (UCI) with $515,345$ instances and 90 features. We use the original train/test splits. (ii) The *US flight delay (Airline)* dataset (Dutordoir et al., 2020). Following Ortega et al. (2023) we use the first $700,000$ instances for training and the next $100,000$ for testing. 8 features are considered: month, day of month, day of week, plane age, air time, distance, arrival time and departure time. (iii) The *Taxi dataset*, with data recorded on January, 2023 (Salimbeni and Deisenroth, 2017). 9 attributes are considered: time of day, day of week, day of month, month, PULocationID, DOLocationID, distance and duration. We filter trips shorter than 10 seconds and larger than 5 hours, resulting in 3 million instances. The first 80% is used as train data, the next 10% as validation data, and the last (10%) as test data. In all experiments, a 3-layer DNN with 200 units, *tanh* activations and L2 regularization is considered. VaLLA and ELLA use 100 inducing points and 100 random points, respectively.

The table displayed on the l.h.s. of Figure 4 presents the averaged results over 5 random seeds. LLA is not considered here due to intractability. Negative log likelihood (NLL), continuous ranked probability score (CRPS) (Gneiting and Raftery, 2007) and a centered quantile metric (CQM), described below, are reported. We observe that VaLLA performs best according to NLL and CQM, while it gives worse results in terms of CRPS compared to the other methods.

| | Airline | | | Year | | | Taxi | | |
|---|---|---|---|---|---|---|---|---|---|
| Model | NLL | CRPS | CQM | NLL | CRPS | CQM | NLL | CRPS | CQM |
| MAP | 5.087 | 18.436 | 0.158 | 3.674 | 5.056 | 0.164 | 3.763 | **3.753** | **0.227** |
| LLA Diag | 5.096 | **18.317** | 0.144 | 3.650 | 4.957 | 0.122 | 3.714 | 3.979 | 0.270 |
| LLA KFAC | 5.097 | **18.317** | 0.144 | 3.650 | **4.955** | 0.121 | 3.705 | 3.977 | 0.270 |
| LLA* | 5.097 | **18.319** | 0.144 | 3.650 | **4.954** | 0.120 | 3.718 | 3.975 | 0.270 |
| LLA* KFAC | 5.097 | **18.317** | 0.144 | 3.650 | **4.954** | 0.120 | 3.718 | 3.976 | 0.270 |
| ELLA | 5.086 | 18.437 | 0.158 | 3.674 | 5.056 | 0.164 | 3.753 | **3.754** | **0.227** |
| VaLLA 100 | **4.923** | 18.610 | **0.109** | **3.527** | 5.071 | **0.084** | **3.287** | 3.968 | **0.188** |
| VaLLA 200 | **4.918** | 18.615 | **0.107** | **3.493** | 5.026 | **0.076** | **3.280** | 3.993 | **0.188** |

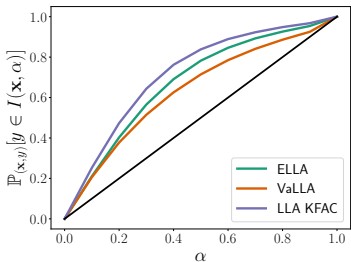

Figure 4: (left) Results on regression datasets. (right) Illustration of CQM on Taxi. Average results across 5 different random seeds (standard deviations always $< 10^{-4}$ and omitted). Best value highlighted in **purple** and second to best in **teal**. * for Last Layer LLA.

**Centered Quantile Metric (CQM).** In regression, CQM assesses the calibration of the predictions, extending the Expected Calibration Error (ECE) to regression problems with Gaussian predictions *with the same mean* but different variance. CQM calculates the *centered interval around the mean* with probability mass $\alpha \in (0,1)$. For Gaussian predictions $\mathcal{N}(\mu(\mathbf{x}), \sigma^2(\mathbf{x}))$, the open interval is defined as $I(\mathbf{x}, \alpha) = (\lambda(-\alpha), \lambda(\alpha))$, where $\lambda(\alpha) = \Phi_{\mu(\mathbf{x}),\sigma^2(\mathbf{x})}^{-1}(\frac{1+\alpha}{2})$ with $\Phi_{\mu(\mathbf{x}),\sigma^2(\mathbf{x})}$ the CDF of a Gaussian with mean $\mu(\mathbf{x})$ and variance $\sigma^2(\mathbf{x})$. The fraction $\gamma$ of test points falling inside the interval is then computed. If the predictive distribution is well calibrated, $\gamma \approx \alpha$. Formally,

$$\text{CQM} = \int_0^1 \left| \mathbb{P}_{(\mathbf{x}^\star, y^\star)} \left[ y^\star \in I(\mathbf{x}^\star, \alpha) \right] - \alpha \right| d\alpha . \tag{51}$$

All methods utilize the pre-trained DNN solution as predictive mean. Thus, the differences in $I(\mathbf{x}, \alpha)$ stem from the predictive variance. Evaluating the integrand in (51) on a grid of $\alpha$ values allows us to visually interpret the uncertainty estimation of each method. The r.h.s. of Figure 4 shows $\mathbb{P}_{(\mathbf{x}, y)} \left[ y \in I(\mathbf{x}, \alpha) \right]$ for several models on the Taxi dataset. In general, all methods tend to over-estimate the actual predictive variance, as evidenced by the values above the diagonal. The l.h.s. of Figure 4 shows CQM estimated using trapezoid integration with 11 points. We refer to Appendix I for more details on the CQM metric.

### G.3. Image Classification Problems

**MNIST and FMNIST.** We employ a 2-layer fully connected DNN with 200 units in each layer and *tanh* activations. In VaLLA we considered 100 and 200 inducing points, while in ELLA, 2000 random points are used. The Out-of-distribution (OOD) detection ability of each method is evaluated using the entropy of the predictive distribution as a score. We compute the area under the ROC curve (AUC) of the binary problem that distinguishes between instances from pairs of datasets MNIST/FMNIST and FMNIST/MNIST (Immer et al., 2021). Moreover, in FMNIST we also assess the robustness of the predictive distribution by rotating the test images up to 180 degrees and computing the ECE and NLL on rotated images (Ovadia et al., 2019).

| Model | ACC | NLL | ECE | BRIER | OOD-AUC |
|---|---|---|---|---|---|
| MAP | **97.6** | **0.076** | **0.008** | **0.036** | 0.905 |
| LLA Diag | 97.4 | 0.143 | 0.072 | 0.053 | 0.922 |
| LLA KFAC | 97.5 | 0.094 | 0.029 | 0.041 | **0.949** |
| LLA* | **97.6** | 0.081 | 0.015 | 0.037 | 0.909 |
| LLA* KFAC | **97.6** | 0.081 | 0.015 | 0.037 | 0.909 |
| ELLA | **97.6** | **0.076** | **0.008** | **0.036** | 0.905 |
| Sampled LLA | **97.6** | 0.087 | 0.026 | 0.040 | **0.954** |
| VaLLA 100 | **97.7** | **0.076** | **0.010** | **0.036** | 0.916 |
| VaLLA 200 | **97.7** | **0.075** | **0.010** | **0.035** | 0.921 |

Figure 5: (left) MNIST experiments. Results averaged over 5 different random seeds (standard deviations $< 10^{-4}$ in all cases and omitted). (right) Box-plots of training times in seconds. ELLA considers 10 prior values chosen using a validation set. Sampled-LLA uses 8 EM steps and 32 samples. Best value is highlighted in **purple** and second to best in **teal**. * for Last Layer LLA.

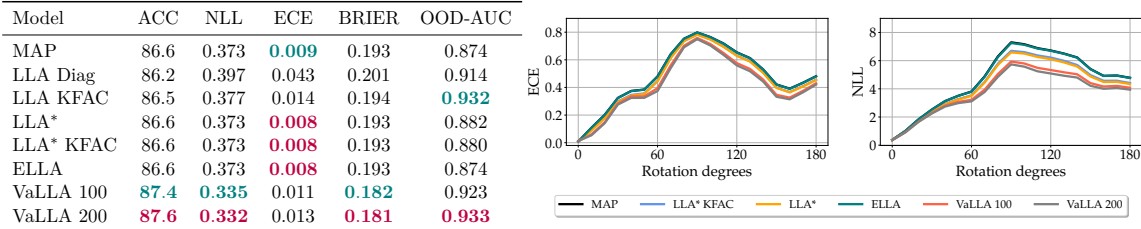

| Model | ACC | NLL | ECE | BRIER | OOD-AUC |
|---|---|---|---|---|---|
| MAP | 86.6 | 0.373 | **0.009** | 0.193 | 0.874 |
| LLA Diag | 86.2 | 0.397 | 0.043 | 0.201 | 0.914 |
| LLA KFAC | 86.5 | 0.377 | 0.014 | 0.194 | **0.932** |
| LLA* | 86.6 | 0.373 | **0.008** | 0.193 | 0.882 |
| LLA* KFAC | 86.6 | 0.373 | **0.008** | 0.193 | 0.880 |
| ELLA | 86.6 | 0.373 | **0.008** | 0.193 | 0.874 |
| VaLLA 100 | **87.4** | **0.335** | 0.011 | **0.182** | 0.923 |
| VaLLA 200 | **87.6** | **0.332** | 0.013 | **0.181** | **0.933** |

Figure 6: (left) Results on FMNIST. Results are averaged over 5 different random seeds (standard deviations are lower than $10^{-4}$ and omitted). Best value is highlighted in **purple** and second to best in **teal**. * for Last Layer LLA. (right) ECE and NLL for rotated FMNIST.

The left table in Figure 5 shows the results on MNIST. VaLLA gives better uncertainty estimates in terms of NLL and the Brier score, but performs less effectively in terms of ECE. Remarkably, VaLLA improves prediction accuracy (ACC) due to the approximation of Daxberger et al. (2021b) to compute class probabilities in multi-class problems. In terms of OOD-AUC VaLLA outperforms the MAP solution but lags behind other methods *s.a.* Sampled-LLA or LLA with Kronecker approximations. Figure 5 (right) illustrates the training times for each method, with VaLLA being faster than ELLA, Sampled-LLA or Last-Layer LLA.

Finally, the left table in Figure 6 displays the results on FMNIST. Here, VaLLA excels in prediction accuracy and provides the best uncertainty estimates in terms of NLL and the Brier score. Although it does not perform as well in terms of ECE, the differences are small. VaLLA also achieves the best results in OOD-AUC. Figure 6 (right) shows VaLLA holds better performance in terms of ECE and NLL as the test images' corruption increases (rotation level), indicating the greater robustness of VaLLA's predictive distribution.

**CIFAR10 and ResNet.** Among with Table 1 that shows ACC, NLL and ECE in the ResNet-Cifar10 experiment, Figure 7 shows the NLL on the perturbed test set with 5

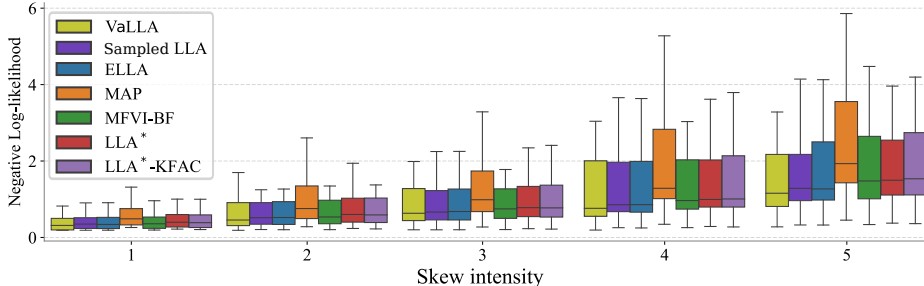

Figure 7: Results on corrupted CIFAR10 with ResNet56. Sampled LLA uses 64 samples and ELLA uses $M = 2000$ and $K = 20$.

increasing levels of 19 image corruptions (Deng et al., 2022). Each box-plot summarizes the test NLL for each intensity level across all 19 corruptions. The results again highlight VaLLA's robust predictive distribution, achieving also lower NLL compared to the other methods.

## Appendix H. Experimental Details

Source code for the conducted experiments can be accessed in the following anonymous repository: https://github.com/Ludvins/Variational-LLA/tree/main.

### H.1. MAP solutions

For regression problems (Year, Airline and Taxi datasets), a 3-layer fully connected NN was used with 200 units in each layer. The optimal weights are obtained by minimizing the RMSE using 20000 iterations of batch size 100 and Adam optimizer (Kingma and Ba, 2015) with learning rate $10^{-2}$ and weight decay $10^{-2}$.

For MNIST and FMNIST experiments, a 2-layer fully connected NN was used with 200 units in each layer. The optimal weights are obtained by minimizing the NLL using 20000 iterations of batch size 100 and Adam optimizer (Kingma and Ba, 2015) with learning rate $10^{-3}$ and weight decay $10^{-3}$.

### H.2. Laplace Library

The Laplace library (Daxberger et al., 2021a) was used to perform last-layer, KFAC and diagonal approximations of the LLA method and optimize the prior variance on each case. The latter is done by optimizing the log marginal likelihood of the data using the library's `log_marginal_likelihood` method for 40.000 iterations with the Adam optimizer and learning rate $10^{-3}$.

### H.3. Efficient Kernel Computation for MLP

In this section we discuss an efficient implementation for computing the Neural Tangent Kernel $\kappa(\mathbf{x}, \mathbf{x}')$. First of all, take into account that the computation of the kernel can be

reduced to a summation on the number of parameters of the model:

$$\kappa(\mathbf{x}, \mathbf{x}') = \sigma_0^2 J_{\hat{\boldsymbol{\theta}}}(\mathbf{x})^T J_{\hat{\boldsymbol{\theta}}}(\mathbf{x}') = \sigma_0^2 \sum_{\theta_s \in \hat{\boldsymbol{\theta}}} \frac{\partial}{\partial \theta_s} g(\mathbf{x}, \hat{\boldsymbol{\theta}}) \frac{\partial}{\partial \theta_s} g(\mathbf{x}', \hat{\boldsymbol{\theta}}). \tag{52}$$

One of the limitations of computing the kernel is storing $J_{\hat{\boldsymbol{\theta}}}(\mathbf{x})$ in memory, which is a 3 dimensional tensor of (batch size, number of classes, number of parameters). Computing the kernel as a sum allows to simplify the required computations significantly (we no longer have to store in memory the Jacobians). Consider now a MLP as

$$g(\mathbf{x}, \hat{\boldsymbol{\theta}}) = h_L \circ a \circ H_{L-1} \circ \cdots \circ a \circ h_1(\mathbf{x}), \tag{53}$$

where each function $a$ is a non-linear activation function and each function $h$ is a linear function of the form

$$h_l(\mathbf{x}) = \boldsymbol{W}_l^T \mathbf{x} + \boldsymbol{b}_l. \tag{54}$$

With this, $g$ is supposed to be a fully-connected neural network of $L$ layers. Each of the partial derivatives of the neural network are

$$\frac{\partial}{\partial W_{l,j,i}} g(\mathbf{x}, \hat{\boldsymbol{\theta}}) \quad \text{and} \quad \frac{\partial}{\partial b_{l,j}} g(\mathbf{x}, \hat{\boldsymbol{\theta}}) \quad \forall l = 1, \ldots, L, \tag{55}$$

and the kernel is computed simply by adding the product of these derivatives. Here, $i$ is a sub-index denoting input $i$-th to layer $l$. Similarly, $j$ is a sub-index denoting each component of the bias vector parameter at layer $l$, or similarly, each output of that layer.

In fact, using the structure of the model and the chain rule, the derivative of the $o^{th}$ output of the network w.r.t. the $j^{th}, i^{th}$ weight parameter of the $l^{th}$ layer is:

$$\frac{\partial}{\partial W_{l,j,i}} g_o(\mathbf{x}, \hat{\boldsymbol{\theta}}) = \left( \frac{\partial}{\partial h_l} g_o(\mathbf{x}, \hat{\boldsymbol{\theta}}) \right)^T \left( \frac{\partial}{\partial W_{l,j,i}} h_l \right), \tag{56}$$

where each of the two vectors in the r.h.s. has length equal to the number of units in the layer $l$. In fact

$$\frac{\partial}{\partial W_{l,j,i}} h_l = \mathbf{1}_l \cdot a(h_{l-1})_i, \tag{57}$$

where $a(h_{l-1})_i$ corresponds to the inputs of the $l^{th}$ layer. Moreover, $\frac{\partial}{\partial h_l} g_o(\mathbf{x}, \hat{\boldsymbol{\theta}})$ can also be computed using the chain rule:

$$\frac{\partial}{\partial h_l} g_o(\mathbf{x}, \hat{\boldsymbol{\theta}}) = \frac{\partial}{\partial h_{l+1}} g_o(\mathbf{x}, \hat{\boldsymbol{\theta}}) \frac{\partial}{\partial h_l} h_{l+1} = \frac{\partial}{\partial h_{l+1}} g_o(\mathbf{x}, \hat{\boldsymbol{\theta}}) \boldsymbol{W}_l^T \text{diag}(a'(h_l)), \tag{58}$$

which can be easily computed by back-propagating the derivatives. The same derivations can be easily done for the biases of each layer $b_{l,j}$. As a result, the derivatives only depend on a back-propagating term $\frac{\partial}{\partial h_l} g_o(\mathbf{x}, \hat{\boldsymbol{\theta}})$ for each layer, the value of the parameters $\boldsymbol{W}_l, \boldsymbol{b}_l$ and the propagated outputs at each layer $h_1, \ldots, h_{L-1}$ evaluated at the non-linear activation $a(\cdot)$ and its derivative $a'(\cdot)$. This means that, if we store the intermediate outputs of each layer $(h_1, \ldots, h_{L-1})$ on the forward pass of the model, by using a single backward pass, we can compute $\frac{\partial}{\partial h_l} g_o(\mathbf{x}, \hat{\boldsymbol{\theta}})$ for each layer.

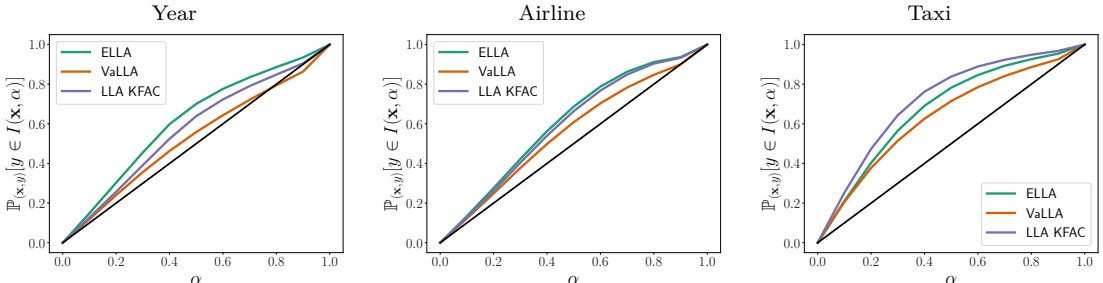

Figure 8: Illustration of CQM, for each method, on the regression datasets Year (left), Airline (middle) and Taxi (right). The MAP solution is given by a fully connected network with 3 hidden layers of 200 units and *tanh* activations.

Critically, given each $\frac{\partial}{\partial h_l} g_o(\mathbf{x}, \hat{\boldsymbol{\theta}})$, we can add the contribution of each layer to the kernel, using (56). In this process, we can sped-up the computations by using structure in the derivatives. For example, in (57) we observe that the derivative has a simple form which is a vector of ones times a scalar. Furthermore, there is no dependence on $j$, the output unit corresponding to the weight $W_{l,j,i}$. Therefore, for two instances $\mathbf{x}$ and $\mathbf{x}'$, the kernel contribution (ignoring the prior variance parameter) corresponding to outputs $o$ and $o'$ is:

$$
\begin{aligned}
\frac{\partial}{\partial W_{l,j,i}} g_o(\mathbf{x}, \hat{\boldsymbol{\theta}}) \frac{\partial}{\partial W_{l,j,i}} g'_o(\mathbf{x}', \hat{\boldsymbol{\theta}}) &= \left( \frac{\partial}{\partial h_l} g_o(\mathbf{x}, \hat{\boldsymbol{\theta}}) \right)^T \left( \frac{\partial}{\partial W_{l,j,i}} h_l \right) \left( \frac{\partial}{\partial h_l} g_o(\mathbf{x}', \hat{\boldsymbol{\theta}}) \right)^T \left( \frac{\partial}{\partial W_{l,j,i}} h_l \right) \\
&= \left( \frac{\partial}{\partial h_l} g_o(\mathbf{x}, \hat{\boldsymbol{\theta}}) \right)^T \left( \frac{\partial}{\partial W_{l,j,i}} h_l \right) \left( \frac{\partial}{\partial W_{l,j,i}} h_l \right)^T \left( \frac{\partial}{\partial h_l} g'_o(\mathbf{x}', \hat{\boldsymbol{\theta}}) \right) \\
&= \left( \frac{\partial}{\partial h_l} g_o(\mathbf{x}, \hat{\boldsymbol{\theta}}) \right)^T \mathbf{1}_l \cdot a(h_{l-1}(\mathbf{x}))_i \cdot a(h_{l-1}(\mathbf{x}'))_i \mathbf{1}_l^T \left( \frac{\partial}{\partial h_l} g'_o(\mathbf{x}', \hat{\boldsymbol{\theta}}) \right) \\
&= s_{o,\mathbf{x}}^l a(h_{l-1}(\mathbf{x}))_i \cdot a(h_{l-1}(\mathbf{x}'))_i s_{o',\mathbf{x}'}^l , \\
&= s_{o,\mathbf{x}}^l \mathbf{s}_{o',\mathbf{x}'}^l a(h_{l-1}(\mathbf{x}))_i a(h_{l-1}(\mathbf{x}'))_i ,
\end{aligned}
\tag{59}
$$

with $s_{o,\mathbf{x}}^l = \left( \frac{\partial}{\partial h_l} g_o(\mathbf{x}, \hat{\boldsymbol{\theta}}) \right)^T \mathbf{1}_l$ a scalar. Similar simplifications occur in the case of, *e.g.*, a convolutional layer.

Summing up, by using this method, all the required kernel matrices can be easily and efficiently computed, for a mini-batch of data points and a set of inducing points, with a similar cost as that of letting the mini-batch or the inducing points go through the DNN. A disadvantage is, however, that the described computations will have to be manually coded for each different DNN architecture. This becomes tedious in the case of very big DNN with complicated layers, as described in Section 3.2.

## Appendix I. Further Analysis of the Quantile Metric

As stated in Section 4, we proposed a new metric for regression problems that, in a way, extends ECE to regression problems with Gaussian predictive distributions *with the same*

*mean.* This kind of metric is desirable for LLA methods as all of them rely on keeping the optimal MAP solution as the predictive mean of the model. They only differ in the predictive variance. Formally, CQM computes for each $\alpha \in (0, 1)$ the probability that points fall into the predictive centered interval of probability $\alpha$. The underlying reasoning is that, if the model explains the data well enough, $\alpha \cdot 100\%$ of the points will fall inside the $\alpha \cdot 100\%$ centered quantile interval. Thus, the metric defined as

$$\mathrm{CQM} = \int_0^1 \left| \mathbb{P}_{(\mathbf{x}^\star, y^\star)} \left[ y^\star \in I(\mathbf{x}^\star, \alpha) \right] - \alpha \right| d\alpha , \tag{60}$$

should be roughly 0 when the model predictive distribution is similar to the actual one, given by the observed data.

Figure 8 shows the evolution of $\mathbb{P}_{(\mathbf{x}^\star, y^\star)} \left[ y^\star \in I(\mathbf{x}^\star, \alpha) \right]$ w.r.t. $\alpha$ for the best performing models in the regression problems. CQM corresponds to the area between the shown curve and $y = x$ (black line). This figure allows to argue that (in general) all methods are over-estimating the predictive variance as they are giving values above the diagonal. That is, for a specific value of $\alpha \in (0, 1)$, the reported probabilities are higher than $\alpha$, meaning that, on average, there are more points in $I(\mathbf{x}, \alpha)$ than they should. That is, the predicted interval is larger than it should, which can only mean that the variance is over-estimated. From a geometrical perspective, it is clear that CQM is always greater than 0 and lower than 0.5; independently of the model and dataset used.

In fact, this figure allow to visually study the level of over/infra-estimation of the prediction uncertainty, for each degree of confidence $\alpha$. For example, in the Year dataset (Figure 8) we see that VaLLA slightly over-estimates the uncertainty for $\alpha \in (0, 0.7)$ while it infra-estimates it for larger values of $\alpha$.

