# OpenReview forum: "Variational Linearized Laplace Approximation for Bayesian Deep Learning"
_approximateinference.org/AABI/2024/Symposium — AABI 2024_

### Official Review · Reviewer_FL2J · 2024-04-10

**Rating:** 6
**Confidence:** 2

**Review:**

**Paper summary**

The paper proposes variational LLA to improve the calibration of deep networks for both regression and classification problems. The idea is to use sparse GP approximation as the variational Laplace posterior of the network. This is achieved by decoupling the mean's inducing points of the GP from its kernel's inducing points and optimizing the location and values of the kernel's inducing points while holding the mean's inducing points constant. The optimized covariance matrix minimizing a variational cost is shown to have a certain form defined by the network's (scaled) neural tangent kernel. Comparisons are shown on multiple toy and real regression and classification datasets where the proposed VaLLA model seems to outperform alternative Laplace approximation methods on (in-distribution) calibration metrics. Some results on out-of-distribution calibration are shown in the appendix where VaLLA seems to be on par with other best-performing methods.

**Strengths**

* The presentation of the arguments follows a logical order, given the small number of pages devoted to the main manuscript.

* The method is sensible and seems to perform well at least for in-distribution calibration.

* The limitation section of the paper does a good job in describing paper's shortcomings and directions for future research.

**Weaknesses**

* An important motivation of model calibration is for out-of-distribution data as the authors allude to it in the introduction. The results mostly show improvement over other methods for in-distribution calibration. The

* It's unclear under what conditions Laplace approximation to the posterior is a reasonable approximation. Specifically, it's argued in the literature that neural networks have multi-modal posteriors and a uni-modal approximation can hurt the calibration and uncertainty estimation. While this paper builds on the literature on Laplace approximation for deep networks, it's still unclear if the proposed model results in any improvement for more complex posterior distributions.

* There's little explanation of the differences between the compared models. The introduction could elaborate more on each of these methods and how they connect to the presented method. I didn't know how LLA is different from LLA* and what's LLA KFAC before going through the Daxerger et. al. paper. Specifically, I was interested in knowing if prior LLA models can be cast as a special case of the VaLLA for specific choices of inducing points.

* Can the authors include GP or SGMCMC in their comparisons as the gold standard baseline?

* As the authors explain, the method seems to have a poor run time, which undermines the primary purpose of Laplace approximation for performing fast and scalable calibration. While the authors include one run-time comparison in Fig. 5 of the appendix a more systematic comparison between different methods seems to be missing from the paper.

* When comparing different methods in the appendix on real datasets, it looks like LLA KFAC does a pretty good job. Can the authors elaborate on potential causes?

---

### Official Review · Reviewer_iT15 · 2024-04-24
**An excellent contribution to the growing Laplace approximation literature on NNs**

**Rating:** 9
**Confidence:** 5

**Review:**

The authors tackle the computational bottleneck of linearized Laplace approximations in neural networks through links to Gaussian processes. In particular, they leverage the dual representation in the RKHS of sparse GPs to scale Laplace approximations.

Question:
Could the authors relate their work to the recent ICLR paper on "Function-space Parameterization of Neural Networks for Sequential Learning". The idea seems related.

---

### Official Review · Reviewer_F3Lp · 2024-04-24
**Interesting approximation of GPs**

**Rating:** 7
**Confidence:** 3

**Review:**

Bayesian algorithms to train deep neural networks (DNN) rely on approximations (Variational approximations, Laplace, etc). It is known that a linear Laplace approximation (LLA) of a DNN leads to a Gaussian process posterior (Khan et al, NeurIPS, 2019). This leads to computational problems as the evaluation of the mean and variance of this GP require matrix inversions that are only feasible to small a sample size N.

The authors propose a variational approximation of the LLA (VaLLA) by a sparse Gaussian process (GP), parametrized by M<<N inducing points. This leads to a provable gain in computational efficiency. The authors prove that the approximation by a sparse GP can be arbitrarily accurate (at the cost of a larger M). They provide comparison to other GP approximations, with promising results.

Strengths:

- Efficient stochastic approximation algorithms can be used to optimize the approximations.
- Promising experimental results, competitive with existing methods.
- Honnest discussion of the limitations ot the method.

Weaknesses:

- While the fundations of the method are well discussed, I would appreciate to see the whole algorithms written clearly in the appendix. How the optimization with respect to the inducing points work is not clear to me.
- Limitations pointed out by the authors: the method becomes infeasible when M becomes large, which might be in some cases a limitation in terms of accuracy.

Recommendation:

Overall, I think this is a good contribution to the AABI 2024 Workshop track.

---

### Official Review · Reviewer_Jb25 · 2024-04-24
**Evaluation could benefit from timings**

**Rating:** 6
**Confidence:** 3

**Review:**

The authors propose a more computationally efficient alternative to generating uncertainty intervals in the linearised Laplace approximation to Bayesian neural network posterior predictive distributions. The linearised Laplace approximation produces a Gaussian approximation to the posterior. Thinking about the neural network as a function, this can be interpreted as a Gaussian process posterior. The modification in this work uses a sparse variational approximation to that Gaussian process posterior. In order to not compromise on predictive mean, the approximation is only used for the variance.

To my knowledge the work is original, though quite incremental. The evaluation is fairly thorough, with a good amount of extra experimental results in the appendix. Uncertainty estimated for neural network predictions is of significant interest, and this work attempts to make one method for generating those estimates faster, which is an important challenge.

I did not check the details of the proofs so cannot guarantee the correctness of the theoretical statements in the main text. It's not clear to me that the assumption of Proposition 2 ($g$ is in $\mathcal{H}$) is satisfied in practice; the authors should comment on this.

The paper is fairly readable, though see minor points at the end of the review.

The major weakness of this work is that the experimental evaluation does not include timing results. The main issue this work addresses is that LLA is computationally expensive; it would be helpful to get a quantitative idea of how the various approximations of Table 1 compare in timings.

Minor points:
* Abstract -- GGN not defined until main text
* Section 1 paragraph 2 -- it should be made clear that the posterior mode is only approximately identified, and the mode found may be only a local maximum in the posterior.
* Same paragraph -- "LA's counterpart". Not sure what is meant by this. From the context it looks like the authors are referring to a drawback or limitation.
* Figure 1 -- The VaLLA panel needs a bigger legend and more clearly different markers for the initial and optimised inducing points.

---

### Official Review · Reviewer_FGgX · 2024-04-27
**solid method, with an interesting, thorough, and clear presentation**

**Rating:** 8
**Confidence:** 3

**Review:**

## Quality
Very high: lots of precise details included about the background, mathematics, algorithms, and experiments. Thorough motivations along with explanation of costs and limitations. Strong baseline comparisons in experiments.

## Clarity
High: it's a complex subject, but all the material is summarized concisely. The motivation is clearly stated. I especially appreciated the crystal clear discussion of limitations in Sec 3.2. One thing that could've improved the clarity would be a summary of exactly what the algorithms do and produce after the math all gets out of the way, e.g. in the form of pseudocode. (Maybe that is contained in the appendices in some form.)

## Originality
Good: the basic idea, of using a variational sparse GP to approximate a Laplace approximation to a DNN (for computational savings) might be clear to some experts in the field (not me!), but there's additional originality in the identification and fixing of problems here (e.g. keeping the predictive mean at the pre-trained DNN output).

## Significance
High: methods that improve Bayesian deep learning, especially by minimizing computational costs and improving scalability while not impeding DNN accuracy, seem like the holy grail right now. This paper directly attacks that problem. Beyond introducing a particular method, the general idea of applying GP techniques to linearized Laplace approximations could be a useful direction for further research (and as far as I can tell it's novel in this paper, though correct me if I'm wrong).

## Strengths
* strong method proposed for general Bayesian deep learning
* clear and thorough presentation, despite the complex topic
* great discussion of limitations and costs
* strong baselines in experiments

## Weaknesses
* there's significant complexity here, which could limit its reach and audience; maybe some of that could be tamed by summary pseudocode

## Questions
* Does it make sense to apply these techniques to finetuning settings for very large models, e.g. for inference over LoRAs on LLMs?
* In Section 1, "LA's counterpart is the necessity of computing the Hessian at the MAP", do you mean "LA's weakness" or "LA's cost" instead of "LA's counterpart"?

---

### Meta-Review · Area_Chair_Ygof · 2024-05-12

**Recommendation:** Accept (Poster)
**Confidence:** 5

**Metareview:**

This paper proposes to approximate the LLA (function-space) posterior using a variational spare GP, based on the dual RKHS formulation of GPs (the LLA by itself _is_ a GP). Crucially, this approximation retains the predictive mean of the original NN, thus compatible with the core spirit of the LLA.

All reviewers agree this is a very good paper and thus warrants acceptance.

Nevertheless, the authors should address all the reviewers' comments in the camera-ready version.

---

### Decision · Program_Chairs · 2024-05-27

Accept